# Safe and Efficient: A Primal-Dual Method for Offline Convex CMDPs under Partial Data Coverage

**Haobo Zhang**
ShanghaiTech University
zhanghb2023@shanghaitech.edu.cn

**Xiyue Peng**
ShanghaiTech University
pengyx2024@shanghaitech.edu.cn

**Honghao Wei**
Washington State University
honghao.wei@wsu.edu

**Xin Liu**[*]
ShanghaiTech University
liuxin7@shanghaitech.edu.cn

## Abstract

Offline safe reinforcement learning (RL) aims to find an optimal policy using a pre-collected dataset when data collection is impractical or risky. We propose a novel linear programming (LP) based primal-dual algorithm for convex MDPs that incorporates "uncertainty" parameters to improve data efficiency while requiring only partial data coverage assumption. Our theoretical results achieve a sample complexity of $\mathcal{O}(1/(1-\gamma)\sqrt{n})$ under general function approximation, improving the current state-of-the-art by a factor of $1/(1-\gamma)$, where $n$ is the number of data samples in an offline dataset, and $\gamma$ is the discount factor. The numerical experiments validate our theoretical findings, demonstrating the practical efficacy of our approach in achieving improved safety and learning efficiency in safe offline settings.

## 1 Introduction

Safe reinforcement learning (RL) aims to learn a reward-maximizing policy while satisfying multiple safety constraints, demonstrating its practicality in many real-world applications, such as autonomous driving [Kiran et al., 2021], robotics [Levine et al., 2016], and healthcare [Yu et al., 2021]. In these tasks, certain behaviors may potentially harm the agent or its surroundings, which is crucial for task completion. One way to mathematically characterize safe RL is through Constrained Markov Decision Processes (CMDPs) [Altman, 2021], where safety constraints are incorporated into the problem when optimizing the objective.

Offline RL aims to learn a sequence of actions from a pre-collected dataset to address scenarios where interacting with the environment is risky, expensive, or impractical. Ensuring sample efficiency in offline RL with function approximation typically requires additional assumptions about both the function classes and the dataset due to training instability and distribution shift issues [Fujimoto et al., 2019, Kostrikov et al., 2021, Paine et al., 2020]. Earlier studies [Chen and Jiang, 2019, Liao et al., 2022, Liu et al., 2019, Wang et al., 2019, Zhang et al., 2020b] in offline RL usually require that all functions in the function space are Bellman-complete and that the dataset has full coverage, meaning it covers the state-action distributions induced by **all** policies. This might be a mild and accepted assumption in offline RL without considering safety. However, it is highly unacceptable and impractical in safe offline RL, as it would require the dataset to cover all hazardous state-action pairs induced by **all** dangerous policies. To address the full coverage issue, later works [Chen and Jiang, 2022, Rashidinejad et al., 2021, Uehara and Sun, 2021, Xie et al., 2021, Zhan et al., 2022, Zhu et al.,

---

[*]Corresponding author.

38th Conference on Neural Information Processing Systems (NeurIPS 2024).

Table 1: Comparison of algorithms for offline safe RL with function approximation.

| Algorithm | Convex MDP | Data Coverage | Function Approximation | Sample Complexity |
|---|---|---|---|---|
| CoptiDICE [Lee et al., 2021] | No | **Partial** | **General** | None |
| DPDL [Chen et al., 2022] | No | **Partial** | None | $\mathcal{O}\left(\frac{1}{(1-\gamma)^2\sqrt{n}}\right)$ |
| MBCL [Le et al., 2019] | No | Full | **General** | $\mathcal{O}\left(\frac{1}{(1-\gamma)^5\sqrt{n}}\right)$ |
| PDCA [Hong et al., 2024] | No | Full | **General** | $\mathcal{O}\left(\frac{1}{(1-\gamma)^2\sqrt{n}}\right)$ |
| Ours | **Yes** | **Partial** | **General** | $\mathcal{O}\left(\frac{1}{(1-\gamma)\sqrt{n}}\right)$ |

2023] reduce the assumption to single-policy coverage by using pessimism in the face of uncertainty. Unfortunately, all existing studies [Chen et al., 2022, Hong et al., 2024, Le et al., 2019] in safe offline RL still require coverage for **all** policies.

Beyond traditional offline safe RL, many applications do not fit the standard RL problem [Abel et al., 2021]. There is a substantial body of literature [Geist et al., 2022, Mutti et al., 2023, Zahavy et al., 2021] studying a more general scenario called convex MDPs, where the objective function is modeled as a convex (or concave) utility function instead of linear, as in the standard RL problem. This framework is quite general and captures various learning scenarios, including imitation [Abbeel and Ng, 2004], exploration [Hazan et al., 2019], and more. However, studying convex MDPs introduces additional challenges. In convex MDPs, moving beyond cumulative rewards means that the Bellman equation fails to hold due to the lack of reward additivity. This leads to breakdowns in many techniques based on Dynamic Programming (DP) [Zhang et al., 2020b]. Despite a large body of practical literature [Lee et al., 2021, Xu et al., 2022, Zheng et al., 2023], robust theoretical analysis remains lacking in this setting.

To address the coverage issue and extend the general function approximation setting, we focus on convex MDPs in the safe offline setting. Our main contributions are summarized below (the detailed comparisons can be found in Table 1):

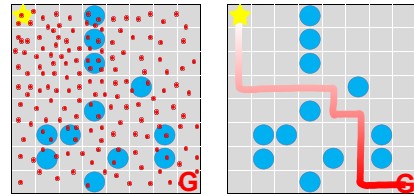

Figure 1: Performance of our algorithm on FrozenLake with completely random data.

• We are the first to study offline convex MDPs under safety constraints with partial data coverage assumption. We reformulate the problem using marginalized importance weights to avoid issues caused by the Bellman equation's failure in convex MDPs.

• We analyze the convergence rate of our proposed approach under partial coverage assumption and theoretically prove that our algorithm achieves $\mathcal{O}\left(\frac{1}{(1-\gamma)\sqrt{n}}\right)$ in both objective and violation bounds with general function approximation, when the number of iteration is larger than $n$. The sample complexity of $\mathcal{O}\left(\frac{1}{(1-\gamma)\sqrt{n}}\right)$ improves the best existing result by a factor of $1/(1-\gamma)^2$.

• Experimental results on Imitation Learning and standard CMDPs demonstrate the generality and effectiveness of our algorithm. As Figure 1 shows our algorithm performs well even with a completely random and safety-violated offline dataset with general function approximation, which verifies our theoretical findings.

## 1.1 Related Work

**Offline Safe RL:** The offline safe reinforcement learning setting entails the agent learning from a fixed dataset while adhering to safety constraints. This involves a blend of offline RL and safe RL, yet research on this approach, particularly concerning theoretical analysis, remains limited. BCQL

---

[2]Note that, they consider standard CMDPs which is a special case of ours.

augmented by BCQ [Fujimoto et al., 2019] optimizes the policy in the offline phase and applies the Lagrange method to handle constraints. CPQ [Xu et al., 2022] tackles the safety constraints by overestimating the cost value function of out-of-distribution ('unsafe') actions and updating the reward value function with 'safe' actions. Another line of work [Lee et al., 2021] using DICE-style technique optimizes policy by calculating the stationary distribution of state-action pairs instead of value function and extracts the policy by importance-sample behavior cloning. CDT [Liu et al., 2023b] combines Decision Transformer with safety constraints and utilizes data augmentation based on Pareto frontiers to enhance the safety and adaptability of the Transformer. It also focuses on the model's capability to different cost thresholds. The work [Le et al., 2019] proposes a meta-algorithm, using Fitted Q Evaluation and Fitted Q Iteration as subroutines to evaluate safety constraints and learn policy respectively. [Chen et al., 2022] analyzes the information-theoretic sample complexity lower bound and proposes a near-optimal primal-dual learning algorithm under partial data coverage but without function approximation. The most related work [Hong et al., 2024] approaches the problem from the perspective of the Actor-Critic algorithm, analyzing the sample complexity to be $\mathcal{O}(\frac{1}{(1-\gamma)^2\sqrt{n}})$ under Slater's condition and full data coverage assumption.

**Convex MDP:** Convex MDP problem extends the scope of MDP by focusing on convex objective functions of stationary distribution, rather than an inner product between reward and stationary distribution. To address the challenge, [Zhang et al., 2020b] introduces a variational Monte Carlo gradient estimation algorithm, demonstrating convergence to the optimal policy across general utility functions. The work [Zahavy et al., 2021] utilizes Fenchel duality to cast convex MDPs as min-max "two-player" games, proposing a meta-algorithm that addresses various convex MDPs through distinct subroutines. The work [Geist et al., 2022] approaches convex MDPs from the Mean-Field Game (MFG) perspective, establishing the equivalence between the optimal condition in convex MDPs and Nash equilibrium in MFGs. [Ying et al., 2023] studies the convex CMDP problem through a policy-based primal-dual algorithm and proves an $\mathcal{O}\left(T^{-1/3}\right)$ convergence rate in both optimality gap and constraint violation. The existing literature on convex MDPs primarily focuses on the online setting, while this paper specifically targets the offline scenario.

## 2 Problem Setup

### 2.1 Convex CMDP problem

We study a discounted constrained Markov decision process (CMDP), denoted by $\mathcal{M} = (\mathcal{S}, \mathcal{A}, R, C, T, \gamma, \mu_0)$, where $\mathcal{S}$ is the state space (a finite set), $\mathcal{A}$ is the action space (a finite set), $R : \mathcal{S} \times \mathcal{A} \to [0, 1]$ and $C : \mathcal{S} \times \mathcal{A} \to [0, 1]$ are reward and cost functions, respectively, $T : \mathcal{S} \times \mathcal{A} \to \Delta(\mathcal{S})$ is the transition probability kernel, where $\Delta(\cdot)$ denotes the probability simplex, $\gamma \in [0, 1)$ denotes the discount factor, and $\mu_0 \in \Delta(\mathcal{S})$ is the initial state distribution. We define a policy $\pi : \mathcal{S} \to \Delta(\mathcal{A})$ as a probability mapping from states to actions. At time slot $t$, the agent observes state $s_t$ and takes action $a_t$ according to the policy $\pi$. For a policy $\pi$, we define its discounted state-action occupancy measure $d_\pi$ as follows

$$d_\pi(s, a) = (1 - \gamma) \sum_{t=0}^{\infty} \gamma^t \mathbb{P}_\pi \left(s_t = s, a_t = a\right), \ \forall s \in \mathcal{S}, \ a \in \mathcal{A},$$

where $\mathbb{P}_\pi \left(s_t, \ a_t\right)$ is the probability of state-action pair $(s, a)$ being visited at time slot $t$ under the policy $\pi$. Further, we let $d_\pi \left(s\right) = \sum_{a \in A} d_\pi \left(s, a\right)$ be the discounted state occupancy measure.

Given the occupancy measure above, we can formulate the following convex CMDP problem

$$\min_{d_\pi \in \mathcal{K}} f\left(d_\pi\right) \quad s.t. \ g\left(d_\pi\right) \leq \tau, \tag{1}$$

where $f$ and $g$ are both convex functions, and $\tau$ is the cost threshold. We present a single safety constraint for simple exposition, and our result can be readily generalized to the setting with multiple constraints. The set $\mathcal{K}$ is a probability simplex that satisfies global balance equations of the underlying Markov process

$$\mathcal{K} = \{d \mid \sum_a d(s, a) = (1 - \gamma)\mu_0(s) + \sum_{s', a'} T\left(s \mid s', a'\right) d\left(s', a'\right), \ \forall s \in \mathcal{S}\}.$$

This set can also be written as a compact matrix form that

$$\mathcal{K} = \{d \mid Md = (1 - \gamma)\mu_0\}, \ M = \text{Diag}(\mathbf{1}_{|\mathcal{A}|}^\top, \cdots, \mathbf{1}_{|\mathcal{A}|}^\top) - \gamma P, \tag{2}$$

where $\mathbf{1}_{|\mathcal{A}|} = [1, 1, \cdots, 1]^\top \in \mathbb{R}^{|\mathcal{A}|}$ and $P \in \mathbb{R}^{|\mathcal{S}| \times |\mathcal{S}||\mathcal{A}|}$ is the transition probability matrix.

The convex CMDP problem in (1) is quite general to capture various learning scenarios, including apprenticeship learning [Abbeel and Ng, 2004], standard CMDPs [Altman, 2021], pure exploration [Hazan et al., 2019], and inverse reinforcement learning in contextual MDPs [Belogolovsky et al., 2021]. Take the standard CMDP as an example, the agent aims to find a policy that maximizes cumulative reward while satisfying the safety constraints [Altman, 2021], which can be written as

$$\max_{d_\pi \in \mathcal{K}} \quad \sum_{s,a} r(s,a) d_\pi(s,a) \quad s.t. \quad \sum_{s,a} c(s,a) d_\pi(s,a) \leq \tau, \tag{3}$$

where $f$ and $g$ are linear functions w.r.t. the state-action occupancy measure.

Take safety-aware apprenticeship learning [Zhou and Li, 2018] for another example, the agent aims to mimic the expert's demonstrations while avoiding the unsafe states in $\mathcal{H}$, which can be written as

$$\min_{d_\pi \in \mathcal{K}} \quad F(d_\pi, d_e) \quad s.t. \quad d_\pi(s) \leq \tau, \ \forall s \in \mathcal{H}. \tag{4}$$

where $d_e$ represents the empirical distribution of the expert's demonstration and $F(\cdot, \cdot)$ is the convex distance function, such as KL-divergence or total variation distance [Zhang et al., 2020a].

However, solving these problems in the online setting can be time-consuming, costly, and potentially dangerous in safety-critical contexts. In contrast, by leveraging historical data, offline RL offers a promising avenue for developing safe and effective algorithms, as introduced next.

## 2.2 Offline Reinforcement Learning

In offline RL, we cannot interact with the environment and only have access to dataset with a finite number of samples. Let $\mathcal{D} = (s_i, a_i, r_i, c_i)_{i=1}^n$ be a collected offline dataset where we assume all the pairs $(s_i, a_i)_{i=1}^n$ are generated independently and identically distributed $(i.i.d.)$ from data distribution $\mu(s,a)$ induced by a behavior policy $\pi_\mu$. Let $n_\mathcal{D}(s,a)$ represents the number of occurrences of the state-action pair $(s,a)$ in the offline dataset $\mathcal{D}$, then $\mu_\mathcal{D}(s,a) = n_\mathcal{D}(s,a)/n$ is an empirical version of $\mu(s,a)$. In offline RL, a major challenge is distribution shift, which measures the mismatch between data distribution and occupancy measure induced by candidate policies. To quantify the distribution shift, we make the following $\pi^*$ concentrability assumption that refers to **partial** data coverage,

**Assumption 1** ($\pi^*$–concentrability). *Let $\pi^*$ be the optimal policy to problem* (1)*, there exists constant $C_{\pi^*} > 0$ such that $d_{\pi^*}(s,a)/\mu(s,a) \leq C_{\pi^*}$, for all $s \in \mathcal{S}, a \in \mathcal{A}$.*

This assumption controls the distribution shift between offline data distribution $\mu$ and the occupancy measure $d_{\pi^*}$ induced by optimal policy $\pi^*$. Specifically, the **partial** data coverage assumption indicates the offline dataset $\mathcal{D}$ should cover state-action pairs visited by the optimal policy $\pi^*$. Unlike the common **full** data coverage assumption in previous works that the dataset $\mathcal{D}$ should encompass data visited by all policies [Chen and Jiang, 2019, Hong et al., 2024, Le et al., 2019], our assumption is considerably more relaxed. Furthermore, in the field of safe RL, **partial** coverage assumption also offers significant advantages, as the **full** coverage implies the behavior policy needs to visit every dangerous state and action space, which is obviously impractical.

Beyond using assumptions to limit distribution shift in offline RL, we consider Marginalized Importance Weight (MIW), a method widely used in the existing literature [Hong et al., 2024, Ozdaglar et al., 2023, Zhan et al., 2022], to further address this challenge.

**Definition 1** (Marginalized Importance Weight). *Given a policy $\pi$, let $d$ be the occupancy measure induced by $\pi$. We define marginalized importance weight $w : \mathcal{S} \times \mathcal{A} \to \mathbb{R}^+$ as $w(s,a) = \frac{d(s,a)}{\mu(s,a)}$, $\forall s \in \mathcal{S}, a \in \mathcal{A}$.*

Note $w$ can be regarded as the density ratio between the normalized discounted occupancy measure and data distribution. Moreover, recall the definition of $w$ and $M$ in equation (2), we define matrix $K \in \mathbb{R}^{|\mathcal{S}| \times |\mathcal{S}||\mathcal{A}|}$ and $K_\mathcal{D} \in \mathbb{R}^{|\mathcal{S}| \times |\mathcal{S}||\mathcal{A}|}$ as

$$K(s', (s,a)) = M(s', (s,a)) \cdot \mu(s,a), \quad K_\mathcal{D}(s', (s,a)) = M(s', (s,a)) \cdot \mu_\mathcal{D}(s,a),$$

where $K_\mathcal{D}$ can be seen as an empirical version of $K$ in dataset $\mathcal{D}$ and it is straightforward to verify $Kw = Md$. With these notations, we establish an equivalent formulation with problem (1) w.r.t.

$$\min_{w \geq 0} \quad f(\mu \cdot w) \tag{5}$$

$$\text{s.t.} \quad Kw = (1-\gamma)\mu_0 \tag{6}$$

$$g(\mu \cdot w) \leq \tau \tag{7}$$

where the operator "$\cdot$" denotes the element-wise product of vectors. As we aim to establish the sample-efficient learning algorithms in large state-action spaces, we regard the importance weight $w$ as a function, i.e. $w : \mathcal{S} \times \mathcal{A} \to \mathbb{R}_+$ that belongs to the convex function class $\mathcal{W}$ in the following assumption.

**Assumption 2** (Realizability). *We assume $w^* \in \mathcal{W}$ where $w^*$ is the optimal solution to the problem (5)–(7).*

This assumption assumes the optimal solution $w^*(\pi^*)$ can be realizable for a convex function class $\mathcal{W}$. Note that the assumption of the convex function class $\mathcal{W}$ is reasonable and standard, which is also achievable with the convexification process in case the function class is non-convex. The convexification process is a common practice in the offline safe RL literature, particularly in the context of general function approximation [Le et al., 2019, Hong et al., 2024]. Now we consider $\mathcal{W}$ as a discrete function class for simplicity and it readily extends to continuous settings (in Remark 1). Further, we introduce a completeness assumption that is used to relax the key constraint in (6).

**Assumption 3** (Completeness). *Let $x_w$ be within the function class $\mathcal{X}$. Define a mapping $\phi$: such that $\phi(w)^\top (Kw - (1-\gamma)\mu_0) = x_w^\top (Kw - (1-\gamma)\mu_0) = \|Kw - (1-\gamma)\mu_0\|_1$. Then, we have $(\mathcal{W}, \mathcal{X})$-completeness under the mapping $\phi$, i.e. $x_w \in \mathcal{X}$ for all $w \in \mathcal{W}$.*

Intuitively, the completeness assumption allows us to replace the computation of the $l_1$-norm with a linear product and simplifies our subsequent analysis. Next, we introduce the standard boundness assumption in offline RL literature with general function approximation [Chen and Jiang, 2019, 2022, Le et al., 2019, Munos and Szepesvári, 2008].

**Assumption 4** (Boundness of $\mathcal{W}$ and $\mathcal{X}$). *We assume function classes $\mathcal{W}$ and $\mathcal{X}$ are bounded, i.e. $\|w\|_\infty \leq B_w$, $\forall w \in \mathcal{W}$ and $\|x_w\|_\infty \leq B_w$, $\forall x_w \in \mathcal{X}$.*

Combining this assumption with Assumption 1 implies $B_w \geq C_{\pi^*}$. Lastly, we impose a mild assumption for the reward and cost functions in the problem (1).

**Assumption 5** (Lipschitz condition). *The functions $f(x)$ and $g(x)$ are convex and satisfy the Lipschitz condition, where there exist constants $L_f$ and $L_g$ such that for any $x, y$, the following inequalities hold $|f(x) - f(y)| \leq L_f \|x - y\|$ and $|g(x) - g(y)| \leq L_g \|x - y\|$.*

## 3 Algorithm Design and Main Results

Despite the practical importance of offline safe RL in real-world applications, there is still a lack of theoretical research on this topic. The earlier literature on this setting often lacks robust theoretical analysis [Lee et al., 2021, Liu et al., 2023b, Xu et al., 2022], and articles with theoretical analysis either yield unsatisfactory results or rely on strong assumptions [Chen et al., 2022, Hong et al., 2024, Le et al., 2019].

In this section, we propose a provable algorithmic framework and establish the first theoretical result in offline convex CMDP, to the best of our knowledge. Moreover, when reducing to standard offline CMDP, we achieve a sample complexity of $\mathcal{O}\left(\frac{1}{(1-\gamma)\sqrt{n}}\right)$ with general function approximation, which improves the current state-of-the-art result by a factor of $1/(1-\gamma)$.

### 3.1 Algorithm Design

Inspired by [Ozdaglar et al., 2023], we first introduce an empirical version of problem (5)–(7) in the offline setting by incorporating a suitable relaxed parameter into the safety constraint. Intuitively, the empirical version of the problem is "close" to the original problem when the dataset is large. This is the key observation for analyzing the convergence performance and safety violations. To solve

(5)–(7), we present its empirical and relaxed problem:

$$\min_{w \in \mathcal{W}} \quad f(\mu_{\mathcal{D}} \cdot w) \tag{8}$$

$$\text{s.t.} \quad \|K_{\mathcal{D}}w - (1-\gamma)\mu_0\|_1 \leq \zeta, \tag{9}$$

$$g(\mu_{\mathcal{D}} \cdot w) - \tau \leq \kappa, \tag{10}$$

where $\mu_{\mathcal{D}}$ and $K_{\mathcal{D}}$ are the empirical version of $\mu$ and $K$, respectively; $\zeta$ and $\kappa$ are the relaxed hyperparameters for the validity constraint (Bellman equation) and safety constraint. Intuitively, the parameters of $\zeta$ and $\kappa$ capture the "uncertainty" induced by the offline dataset $\mathcal{D}$ in terms of distribution shift and safety concerns. The values of these parameters will be specified later and play an important role in our analysis. Next, we demonstrate that the mismatch of the objective and constraint violation bound depends on the "uncertainty", which exhibits $\mathcal{O}(1/\sqrt{n})$ distance.

Note that all parameters in the optimization problem (8)–(10) can be determined from the offline dataset. When the state-action space is not large, this problem can be efficiently solved by convex optimization solvers. However, when the state-action space is large, and the function approximation is necessary (e.g., $w$ is parameterized by a neural network), it would be quite challenging (if not impossible) to solve this problem. To address this challenge, we propose a primal-dual algorithm that is sample-efficient and computationally tractable to solve the problem iteratively. We first introduce the Lagrange function of problem (8)–(10),

$$\mathcal{L}(w, \lambda, \phi) = f(\mu_{\mathcal{D}} \cdot w) + \lambda\left(\|K_{\mathcal{D}}w - (1-\gamma))\mu_0\|_1 - \zeta\right) + \phi\left(g(\mu_{\mathcal{D}} \cdot w) - \tau - \kappa\right), \tag{11}$$

where $\lambda$ and $\phi$ are Lagrange multipliers.

---

**Algorithm 1:** **P**rimal-dual algorithm for **O**ffline **C**onvex **CMDP** (POCC)

1 **Input**: Dataset $\mathcal{D} = \{(s_i, a_i, r_i, c_i)\}_{i=1}^n$, the relaxed parameters $\kappa, \zeta$, and the step size $\eta = \frac{1}{\sqrt{K}}$;

2 **Initialization**: Choose any $w^1 \in \mathcal{W}$ and the Lagrangian multipliers $\lambda^1 = 0, \phi^1 = 0$;

3 **for** $k = 1, 2, \ldots, K$ **do**

4  **Primal:** $w^{k+1} = \mathcal{P}_{\mathcal{W}}\left[w^k - \eta\nabla\mathcal{L}_w(w^k, \lambda^k, \phi^k)\right]$,

   **Dual:** $\phi^{k+1} = \left[\tau^k - \eta\nabla\mathcal{L}_\phi(w^k, \lambda^k, \phi^k)\right]_0^{\phi_{max}^{k+1}}$,  $\lambda^{k+1} = \left[\lambda^k - \eta\nabla\mathcal{L}_\lambda(w^k, \lambda^k, \phi^k)\right]_0^{\lambda_{max}^{k+1}}$,

   where $\mathcal{P}_{\mathcal{W}}$ is the projection onto set $\mathcal{W}$ and $[\cdot]_l^h$ is the projection onto interval $[l, h]$.

5 Compute the average $\overline{w}_K = \sum_{i=1}^K w^i$;

6 Extract the policy $\overline{\pi}_K$ with formula (12);

7 **Output:** Policy $\overline{\pi}_K$ ;

---

Given the Lagrange function above, we introduce our algorithm called POCC (in Algorithm 1), which takes the offline dataset $\mathcal{D}$ as input and runs a primal-dual method on the estimated Lagrange function. Specifically, at each iteration $k$, POCC updates the importance weight by gradient descent and projects it back to function class $\mathcal{W}$, then updates Lagrange multipliers of validity constraint (6) and safety constraint (7) respectively. After $K$ steps, POCC returns an averaged $\overline{w}_K$, we can extract the corresponding policy $\overline{\pi}_K$ based on the offline dataset $\mathcal{D}$ as follows

$$\overline{\pi}_K(a \mid s) := \begin{cases} \frac{\overline{w}_K(s,a)\pi_\mu(a|s)}{\sum_{a' \in \mathcal{A}} \overline{w}_K(s,a')\pi_\mu(a'|s)}, & \text{if } \sum_{a' \in \mathcal{A}} \overline{w}_K(s, a')\pi_\mu(a' \mid s) > 0 \\ \frac{1}{|\mathcal{A}|}, & \text{if } \sum_{a' \in \mathcal{A}} \overline{w}_K(s, a')\pi_\mu(a' \mid s) = 0 \end{cases} \tag{12}$$

where the second equality means that if $\sum_{a' \in \mathcal{A}} \overline{w}_K(s, a')\pi_\mu(a' \mid s) = 0$ we randomly choose an action for state $s$.

### 3.2 Theoretical Results

We present the theoretical results of our proposed approach in the following theorem.

**Theorem 1** (Sample complexity of $\overline{\pi}_K$). *Suppose Assumptions 1–7 hold. Denote $\overline{\pi}_K$ as the corresponding policy induced by $\overline{w}_K$. Set the relaxed parameters $\zeta = \frac{2\sqrt{2}B_w}{\sqrt{n}}\sqrt{\log\frac{|\mathcal{W}||\mathcal{X}|}{\delta}}$ and*

$\kappa = \frac{\sqrt{2}L_g B_w}{\sqrt{n}}\sqrt{\log\frac{2|\mathcal{W}|}{\delta}}$, and the step size $\eta = \frac{1}{\sqrt{K}}$. Let the constants $\upsilon = \frac{1}{1-\gamma}\left(4B + 4L + 2\varepsilon\right)$ and $\iota = \frac{1}{1-\gamma}\left(B^2 + 4B + 4L + \frac{L^2}{\sqrt{K}} + \varepsilon\right)$, we have, with at least $1 - 8\delta$ probability,

$$J_r(\overline{\pi}_K) - J_r(\pi^*) \leq \frac{6L_f B_w \sqrt{2\log(|\mathcal{W}||\mathcal{X}|/\delta)}}{(1-\gamma)\sqrt{n}} + \frac{\iota}{2\sqrt{K}} \tag{13}$$

$$J_c(\overline{\pi}_K) - \tau \leq \frac{6L_g B_w \sqrt{2\log(2|\mathcal{W}||\mathcal{X}|/\delta)}}{(1-\gamma)\sqrt{n}} + \frac{\upsilon}{\sqrt{K}} \tag{14}$$

where $J_r(\overline{\pi}_K) = f(\overline{d}_K)$ and $J_c(\overline{\pi}_K) = g(\overline{d}_K)$ are objective and constraint performance of policy $\pi$ respectively, $B$ represents the distance between initial value $w^1$ of the iteration and optimal solution $w_{\mathcal{D}}$ to problem (8)–(10), $\varepsilon \geq 0$ is a constant, $L$ is the max Lipschitz constant of Lagrange function and $K$ is the number of iterations.

**Remark 1.** *We remark that Theorem 1 remains valid even when the function class $\mathcal{W}$ is a continuous set. In this case, the cardinality $|\mathcal{W}|$ can be replaced with the covering number of $\mathcal{W}$, and the union bound can be applied to its $\epsilon$-covering set. This adjustment also preserves the sample complexity order of $\mathcal{O}(1/\sqrt{n})$, up to a constant dependent on $\epsilon$. For further details on the extension from discrete $\mathcal{W}$ to a continuous set, please refer to [Le et al., 2019, Xie and Jiang, 2021].*

Theorem 1 demonstrates that the convergence performance of our algorithm can be divided into two parts: $\mathcal{O}(1/\sqrt{n}) + \mathcal{O}(1/\sqrt{K})$. Regarding the first term, except for the size of dataset, it mainly depends on the function class (searching space) $\log|\mathcal{W}||\mathcal{X}|$ and relaxed parameters $\zeta$ and $\kappa$ that capture the "uncertainty" for addressing the distribution shift and safety concern. Moreover, the second term $\mathcal{O}(1/\sqrt{K})$ connects with the error bound between $\overline{w}_K$ and optimal solution $w_{\mathcal{D}}$, which decays at the rate of $1/\sqrt{K}$ when the number of iterations increases. It is worth noting that the convergence rate of policy $\overline{\pi}_K$ is still $\mathcal{O}(1/\sqrt{n})$ when the iterative number is sufficient to satisfy $K \geq n$. The algorithm is gradient-based and does not involve additional computations for solving the optimization problem (8)–(10). This suggests that we can improve the convergence performance w.r.t. $K$ by employing more advanced primal-dual techniques to reduce the convergence rate of the second term, such as those discussed in [Yu and Neely, 2017] with a convergence rate of $\mathcal{O}(1/K)$.

Unlike most of the previous work focuses on convex MDPs in online setting [Bai et al., 2023, Ying et al., 2023, Zahavy et al., 2021, Zhang et al., 2020b], Theorem 1 to the best of our knowledge, is the first provable result in offline convex MDPs. Moreover, compared to the best result $\mathcal{O}\left(\frac{1}{(1-\gamma)^2\sqrt{n}}\right)$ in offline CMDP [Hong et al., 2024, Chen et al., 2022], our results achieve a sample complexity of $\mathcal{O}\left(\frac{1}{(1-\gamma)\sqrt{n}}\right)$, which outperform the state-of-the-art by a factor of $1/(1-\gamma)$. Besides, our algorithm is appropriate in the scenario with large-scale state-action space due to the general function approximation for $w$ while the work [Chen et al., 2022] targets the tabular setting; our result is more favorable in the safety applications compared to [Hong et al., 2024] because we replace the realizability of value function $\nu$ with a slightly stronger completeness assumption but reduce the data coverage from **full** to **partial**. As stated in Assumption 1, the **full** data coverage not only implies access to a highly exploratory dataset but also is impractical for offline safe RL, as it assumes behavior policy needs to visit every dangerous state-action pair. Finally, we want to comment that the previous work all focuses on the standard RL, whose objective function is linear, while our algorithm is general enough to tackle the convex MDPs.

**Remark 2.** *In the theorem, we assume prior knowledge of the behavior policy $\pi_\mu$ for the sake of exposition. However, in practice, it is often challenging to know the behavior policy in advance, as we typically only have access to the offline dataset. The most popular approach to tackle this challenge is behavior clone. It posits that $\hat{\pi}_\mu$ can be estimated as $\hat{\pi}_\mu(a|s) = \frac{n(s,a)}{n(s)}$, where $n(s,a)$ denotes the number of the occurrences of the state-action pair $(s,a)$ in the offline dataset. We employed this estimation method in the experiments and results demonstrate its effectiveness.*

## 4 Theoretical Analysis

In this section, we present a sketch of the proof of Theorem 1. We focus on illustrating the analysis of the constraint violation, and the convergence of the objective follows similar steps. The detailed proof can be found in Appendix A.

We first decompose the bound into different major terms and then sutdy them individually

$$J_c(\overline{\pi}_K) - \tau = g(d_{\overline{\pi}_K}) - \tau \tag{15}$$

$$= \underbrace{g(d_{\overline{\pi}_K}) - g(\overline{d}_K)}_{\text{I}} + \underbrace{g(\mu \cdot \overline{w}_K) - g(\mu_{\mathcal{D}} \cdot \overline{w}_K)}_{\text{II}} + \underbrace{g(\mu_{\mathcal{D}} \cdot \overline{w}_K) - \tau}_{\text{III}} \tag{16}$$

The first equality holds due to the definition: $J_c(\overline{\pi}_K) = g(d_{\overline{\pi}_K})$, where $d_{\overline{\pi}_K}$ represents the occupancy measure induced by returned policy $\overline{\pi}_K$. Recall the definition $w(s,a) \cdot \mu(s,a) = d(s,a)$, we then have $g(\overline{d}_K) = g(\mu \cdot \overline{w}_K)$. It is worth noting that the decomposition is intuitive since terms I and II relates to the idea of constructing "uncertainty" parameter for the offline dataset and term III mainly depends on the convergence of primal-dual method.

Specifically, term I is the distance between $g(\overline{d}_K)$ and $g(d_{\overline{\pi}_K})$. It represents the error that we rectify the unnormalized occupancy measure $\overline{d}_K$, which violates the validity constraint (6), to a satisfying one $d_{\overline{\pi}_K}$. The term II illustrates the error incurred when applying the calculated $\overline{w}_K$ from offline dataset $\mathcal{D}$ to the real environment $\mu$, which depends on the sample size of the dataset. The term III is related to the relaxed parameter $\kappa$ in safety constraint (7) and the distance between returned $\overline{w}_K$ and optimal solution $w_{\mathcal{D}}$. Next, we present the following lemmas to bound these terms.

**Lemma 1.** *Suppose Assumptions 1–4 hold, we have, with probability at least $1 - 2\delta$,*

$$J_c(\overline{\pi}_K) - g(\overline{d}_K) \leq \frac{4L_g B_w \sqrt{2\log(|\mathcal{W}||\mathcal{X}|/\delta)}}{(1-\gamma)\sqrt{n}}.$$

**Lemma 2.** *Suppose Assumptions 1–4 hold. For $\overline{w}_K \in \mathcal{W}$, we have, with probability at least $1 - \delta$,*

$$g(\mu \cdot \overline{w}_K) - g(\mu_{\mathcal{D}} \cdot \overline{w}_K) \leq \frac{\sqrt{2}L_g B_w}{\sqrt{n}} \sqrt{\log \frac{2|\mathcal{W}|}{\delta}}.$$

**Lemma 3.** *Suppose Assumptions 1–7 hold. For $\overline{w}_K \in \mathcal{W}$, we have*

$$g(\mu_D \cdot \overline{w}_K) - \tau \leq \kappa + \frac{\upsilon}{\sqrt{K}}.$$

Combining the above lemmas, we have, with at least $1 - 3\delta$ probability,

$$J_c(\overline{\pi}_K) - \tau = \underbrace{g(d_{\overline{\pi}_K}) - g(\overline{d}_K)}_{\text{I}} + \underbrace{g(\mu \cdot \overline{w}_K) - g(\mu_{\mathcal{D}} \cdot \overline{w}_K)}_{\text{II}} + \underbrace{g(\mu_{\mathcal{D}} \cdot \overline{w}_K) - \tau}_{\text{III}}$$

$$\leq \frac{4L_g B_w \sqrt{2\log(|\mathcal{W}||\mathcal{X}|/\delta)}}{(1-\gamma)\sqrt{n}} + \frac{2\sqrt{2}L_g B_w}{\sqrt{n}} \sqrt{\log \frac{2|\mathcal{W}|}{\delta}} + \frac{\upsilon}{\sqrt{K}}$$

$$\leq \frac{6\sqrt{2}L_g B_w \sqrt{2\log(2|\mathcal{W}||\mathcal{X}|/\delta)}}{(1-\gamma)\sqrt{n}} + \frac{\upsilon}{\sqrt{K}}.$$

where we set $\zeta = \frac{2\sqrt{2}B_w}{\sqrt{n}} \sqrt{\log \frac{|\mathcal{W}||\mathcal{X}|}{\delta}}$, $\kappa = \frac{\sqrt{2}L_g B_w}{\sqrt{n}} \sqrt{\log \frac{2|\mathcal{W}|}{\delta}}$.

# 5 Experiments

This section aims to justify the effectiveness of our proposed framework through numerical experiments. We test a practical version of Algorithm 1, where we replace the gradient-type update with the "Adam"-type update (the detailed algorithm can be found in Algorithm 2 in the appendix). We test our algorithm to two specific offline convex CMDPs: 1) Safe imitation learning and 2) standard offline CMDP. Our objective is to address the following questions: (i) Are the experimental results consistent with our theory? (ii) how does the data quality affect the performance of our algorithm? The additional details can be found in the Appendix B.

## 5.1 Safe Imitation Learning

To showcase the generality of our algorithm, we choose imitation learning as a user case of convex MDPs and conduct the experiments in a maze environment. We design the environment, as illustrated

in Figure 2, modified from [Geist et al., 2022]. The problem of (safe) imitation learning can be formulated as $F(d) = KL(d \| d_E)$, where $d_E$ represents the stationary distribution of an expert. The environment is deterministic; agent has four actions (left, down, right, up); moving towards the wall (white) and the boundary does not change the state; the goal is to learn from the expert demonstrations (yellow) under safety constraints.

We collect data by expert demonstrations in (a) but randomly remove 25% states. We intend for the algorithm to learn to fill in the gaps using its inherent properties and function approximation, as emphasized in our theory. We are presenting two sets of results: one that considers safety constraints with a cost threshold of 0, and another that does not consider safety constraints. It's important to note that we cannot simply take the stationary distribution of the dataset as our final result. This is due to several reasons, including the fact that expert demonstrations are incomplete (25% of states are removed), simple replacement leads to poor results, and it's not suitable when considering the safety of the agents.

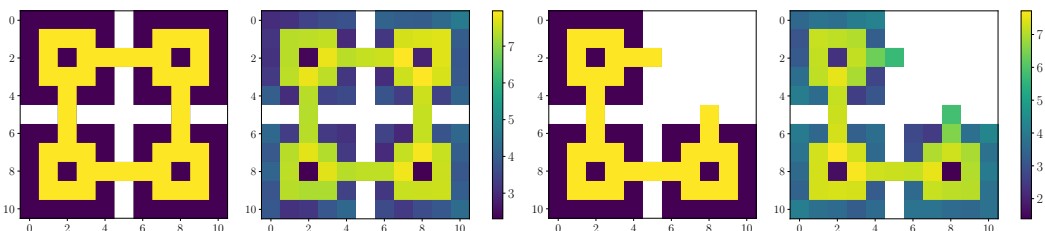

Figure 2: Reading order: **(a)** target demonstrations in yellow, wall in white; **(b)** result for log-density without considering the safety constraint; **(c)** target demonstrations that all states in top-right corners have cost; **(d)** result for log-density with safety constraint.

Our results are presented in Figure 2. In Figure (2b), it is demonstrated that our policy can accurately replicate the expert distribution using function approximation and completely bridge the gap required by the expert. Additionally, when taking into account the safety constraints, which involve setting the cost in the entire top-right corners, it is evident that our policy avoids states with cost and behaves appropriately, aligning with our theoretical framework.

## 5.2 Offline CMDP

We consider an 8x8 grid world environment FrozenLake, with the initial state being the top-left grid. The agent has four actions: N (north), S (south), E (east), and W (west). The primary objective is to reach the goal while avoiding all holes. The game terminates if the agent achieves the goal within 25 steps. A reward of 1 is obtained when the agent achieves the goal, and the main cost function assigns a cost of 1 if the agent falls into a hole and 0 otherwise. The diagram of the environment has been presented in Figure 1 in the introduction.

Initially, we simulate a mixture of different percentages of optimal and uniform policies to collect the offline dataset. We employ various behavior policies $\pi_{\mathcal{D}}$, running 200 trajectories to collect the offline dataset $\mathcal{D}$, with each trajectory having a maximum of 50 time steps to ensure that the optimal goal is included in the dataset.

Additionally, we increase the difficulty of the environment compared to classical FrozenLake, such that if the agent falls into a hole, it can also come out in the next step. This implies that our cost constraint influences the training. We set the cost threshold as 0 here, which means that the agent is not supposed to incur any cost. Note that a higher percentage with a uniform policy indicates that the problem becomes more difficult. We set the discount factor as $\gamma = 0.99$, $\zeta = 0.1$, and $\kappa$ in our algorithm. Furthermore, we encode the environment with one-hot encoding and employ a more practical algorithm. We refer to $w$ as a single hidden-layer neural network which we describe in Algorithm 2 in Appendix B. We set the learning rate of $10^{-5}$ for $w$ and $10^{-4}$ for Lagrange multipliers.

In remark (), we have stated that there are two approaches when facing with the scenario that the behavior policy is unknown, and here we choose to use the behavior clone method that we will estimate the behavior policy through the offline dataset. We compare our algorithm with COptiDICE

[Lee et al., 2021], which is a well-acknowledged baseline algorithm in the offline safe RL literature [Hong et al., 2024, Liu et al., 2023b].

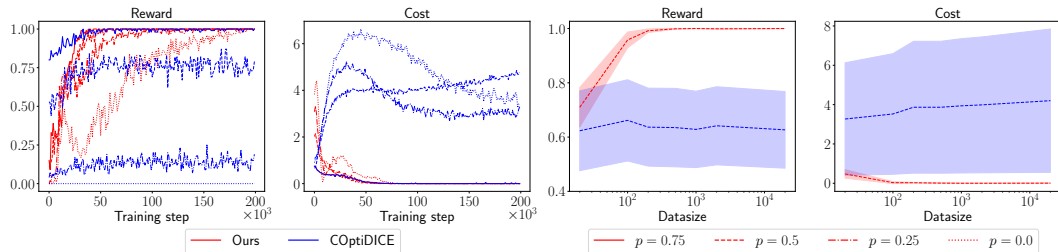

Figure 3: Performance on FrozenLake with general function approximation. Reading order: (a) and (b) show the training result with four different behavior policies of COptiDICE and ours. (c) and (d) demonstrate the variations in rewards and costs as the dataset increases. Each point is the average result of 10 independent runs.

Denote $p$ as the percentage of optimal policy within the behavior policy. We evaluate the algorithm with different behavior policies $p = \{0.75, 0.5, 0.25, 0\}$ and dataset sizes. We present the results in Figure 3. In the first two figures, it can be seen that our algorithm can consistently find the optimal path even with completely random data. Conversely, COptiDICE behaves well when the proportion of optimal policy is 0.75, but cannot even learn a logical and safe policy when the majority of the data in dataset is random and constraint violated. In the last two figures, we test the performance of algorithms in different sizes of dataset, where $p = 0.5$ in behavior policy. The results demonstrate that our algorithm can find a safe and optimal path as the dataset size increases. In contrast, the results from the COptiDICE algorithm show high variance, where it can only find a safe path in about 50% runs.

In summary, our algorithm performs well across various behavior policies and dataset sizes, which is consistent with our theoretical results and assumptions.

## 6  Conclusions

In this work, we consider convex CMDPs in the offline setting. We propose a sample efficient RL approach that addresses the challenges in offline convex CMDPs. We theoretically prove that we can suffer $\mathcal{O}(1/\sqrt{n})$ sample complexity in both performance and violation bound with general function approximation under mild data coverage assumption, which is the first result in offline convex MDPs as best of our knowledge and surpasses the state-of-the-art result in offline safe RL by a factor of $1/(1 - \gamma)$. Experimental studies further demonstrate the effectiveness and generality of our framework.

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

# Appendix

## A   Proof of Theorem 1

In this section, we give the complete proof of Theorem 1. Specifically, we bound the regret first and constraint violation second. Hence, we first present some auxiliary lemmas and combine them to prove the final theorem.

**Proof of objective bound**

We also decompose the expression by adding and subtracting corresponding terms and prove them individually.

$$
\begin{aligned}
J_r(\overline{\pi}_K) - J_r(\pi^*) &= f(d_{\overline{\pi}_K}) - f(d^*) \\
&= f(d_{\overline{\pi}_K}) - f(\overline{d}_K) + f(\overline{d}_K) - f(d^*) \\
&= f(d_{\overline{\pi}_K}) - f(\overline{d}_K) + f(\mu \cdot \overline{w}_K) - f(\mu \cdot w^*) \\
&= \underbrace{f(d_{\overline{\pi}_K}) - f(\overline{d}_K)}_{\text{I}} + \underbrace{f(\mu \cdot \overline{w}_K) - f(\mu_{\mathcal{D}} \cdot \overline{w}_K)}_{\text{II}} + \underbrace{f(\mu_{\mathcal{D}} \cdot \overline{w}_K) - f(\mu \cdot w^*)}_{\text{III}}
\end{aligned}
$$

Next, we will prove these items individually.

**Lemma 4.** $\forall w^* \in \mathcal{W}$, $w^*$ *satisfies constraint functions* (6) *and* (7) *with at least* $1 - 2\delta$ *probability.*

*Proof.* First, we focus on the safety constraint (7). We prove it with Hoeffding inequality. Recall the Hoeffding inequality:

$$
\mathbb{P}\left(\left|\bar{X} - \mathbb{E}[X]\right| \geq t\right) \leq 2\exp\left(-\frac{2n^2t^2}{\sum_{i=1}^n (b_i - a_i)^2}\right)
$$

Take $\mu_{\mathcal{D}} \cdot w$ as the random variable, we have

$$
\mathbb{E}[(\mu_{\mathcal{D}} - \mu) \cdot w] = 0
$$

Since $\mu(s,a) \in [0,1]$, and meanwhile $w$ lies in $[-B_w, B_w]$, we have $(\mu_{\mathcal{D}} - \mu) \cdot w$ lies in interval $[-B_w, B_w]$. To satisfy the safety constraint function:

$$
g(\mu_{\mathcal{D}} \cdot w) - \tau \leq \kappa
$$

We set $t = \frac{\sqrt{2}B_w}{\sqrt{n}}\sqrt{\log\frac{2|\mathcal{W}|}{\delta}}$. Combine with the Hoeffding inequality, we have

$$
\mathbb{P}\left(|\mu_{\mathcal{D}} \cdot w - \mu \cdot w| \geq t\right) \leq \frac{\delta}{|\mathcal{W}|} \tag{17}
$$

For the constraint function, recall we suppose that it satisfies the Lipschitz condition,

$$
|g(\mu_{\mathcal{D}} \cdot w) - g(\mu \cdot w)| \leq L_g \|\mu_{\mathcal{D}} \cdot w - \mu \cdot w\|
$$

Before proceeding to the following derivation, we first state a fact:

$$
\mathbb{P}(z \geq t) \leq \delta \quad \text{and} \quad y \leq z \quad \Longrightarrow \quad \mathbb{P}(y \geq t) \leq \delta
$$

The above equation means that if a variable **z** is greater than $t$ with probability less than or equal to $\delta$, then any variable such as **y** that is lower than that variable must also be lower than $t$ with probability less than or equal to $\delta$.

We get

$$
\mathbb{P}\left(|g(\mu_{\mathcal{D}} \cdot w) - g(\mu \cdot w)| \geq L_g \cdot t\right) \leq \frac{\delta}{|\mathcal{W}|} \tag{18}
$$

Note the fact that $(\mu_{\mathcal{D}} \cdot w - \mu \cdot w)$ is obviously lower than $|\mu_{\mathcal{D}} \cdot w - \mu \cdot w|$, then

$$
\mathbb{P}\left(g(\mu_{\mathcal{D}} \cdot w) - g(\mu \cdot w) \geq L_g \cdot t\right) \leq \frac{\delta}{|\mathcal{W}|} \tag{19}
$$

Take the union bound for $w \in \mathcal{W}$, then

$$\forall w \in \mathcal{W}, \quad \mathbb{P}\left(g(\mu_{\mathcal{D}} \cdot w) - g(\mu \cdot w) \geq L_g \cdot t\right) \leq \delta \tag{20}$$

Then for all $w^* \in \mathcal{W}$,

$$\mathbb{P}\left(g(\mu_{\mathcal{D}} \cdot w^*) - g(\mu \cdot w^*) \geq L_g \cdot t\right) \leq \delta \tag{21}$$

Because $w^*$ is the optimal solution for problem (5)–(7) and we suppose the assumption of realizability is set up, then in equation (7), $w^*$ satisfies

$$g(\mu \cdot w^*) \leq \tau$$

Combing with equation (19), we get

$$\mathbb{P}\left(g(\mu_{\mathcal{D}} \cdot w^*) - \tau \geq L_g \cdot t\right) \leq \delta \tag{22}$$

So we have, with at least $1 - \delta$ probability,

$$g(\mu_{\mathcal{D}} \cdot w^*) - \tau \leq L_g \cdot t = \kappa \tag{23}$$

Next, we focus on the validity constraint function (6).

Here, we take $x^\top (K_D - K)w$ as the random variable and note the fact that it lies in the interval $[-2B_w, 2B_w]$. Use the Hoeffding inequality we have that $\forall x \in B$,

$$\begin{aligned}
\mathbb{P}(x^\top (K_{\mathcal{D}} - K)w \geq t) &\leq \exp\left(-\frac{2n^2 t^2}{\sum_{i=1}^{n}(b_i - a_i)^2}\right) \\
&= \exp\left(\frac{-nt^2}{8B_w^2}\right)
\end{aligned} \tag{24}$$

Set $t = \frac{2B_w \sqrt{2\log(|\mathcal{W}||\mathcal{X}|/\delta)}}{\sqrt{n}}$, the inequality becomes the form as:

$$\mathbb{P}\left(x^\top (K_{\mathcal{D}} - K)w \geq t\right) \leq \frac{\delta}{|\mathcal{W}||\mathcal{X}|} \tag{25}$$

Take the union bound for all $w$ and $x$, we have

$$\mathbb{P}\left(x^\top (K_{\mathcal{D}} - K)w \geq t\right) \leq \delta, \quad \forall w \in \mathcal{W}, \ \forall x \in \mathcal{X} \tag{26}$$

So for all $w^* \in \mathcal{W}$,

$$\mathbb{P}\left(x^\top (K_{\mathcal{D}} - K)w^* \geq t\right) \leq \delta \tag{27}$$

Note the fact $\|Kw^* - (1-\gamma)\mu_0\| = 0$, we have the inequality for all $w^* \in \mathcal{W}$

$$\mathbb{P}\left(x^\top (K_{\mathcal{D}} w^* - (1-\gamma)\mu_0) \geq t\right) \leq \delta \tag{28}$$

That is, with at least $1 - \delta$ probability,

$$x^\top (K_{\mathcal{D}} w^* - (1-\gamma)\mu_0) \leq t = \zeta \tag{29}$$

Take the union bound for equation (23) and (29) completes the proof. $\square$

Next, we show that $f(\mu_{\mathcal{D}} \cdot w_{\mathcal{D}})$ is close to $f(\mu \cdot w^*)$.

Then the empirical covering number $n(\epsilon, \mathcal{W})$ is defined as the size of the smallest $\epsilon$-cover. Intuitively, the $\epsilon$ cover set can represents the original set in the sense of $\epsilon$. This definition is very useful when the original set is continuous and we want to take the union bound of it. By this $\epsilon$ cover we can take the union bound of the continuous set and the distance of them is measured by $\epsilon$.

**Lemma 5.** *We have*

$$f(\mu_{\mathcal{D}} \cdot w_{\mathcal{D}}) \leq f(\mu \cdot w^*) + \frac{\sqrt{2}L_f B_w}{\sqrt{n}}\sqrt{\log \frac{2|\mathcal{W}|}{\delta}}$$

*with $1 - 3\delta$ probability.*

*Proof.* From Lemma 4, we have

$$f\left(\mu_{\mathcal{D}} \cdot w_{\mathcal{D}}\right) \leq f\left(\mu_{\mathcal{D}} \cdot w^*\right)$$

with probability at least $1 - 2\delta$. Now, we use Hoeffding inequality to bound $f\left(\mu_{\mathcal{D}} \cdot w\right) - f\left(\mu \cdot w\right)$. Take $\left(\mu_{\mathcal{D}} - \mu\right) \cdot w$ as the random variable and note it lies in interval $\left[-B_w, \ B_w\right]$, then we have

$$\begin{aligned}
\mathbb{P}\left(\left|\left(\mu_{\mathcal{D}} - \mu\right) \cdot w\right| \geq t\right) &\leq 2\exp\left(-\frac{2n^2 t^2}{\sum_{i=1}^{n}(b_i - a_i)^2}\right) \\
&= 2\exp\left(\frac{-nt^2}{2B_w^2}\right)
\end{aligned} \tag{30}$$

Set $t = \frac{B_w\sqrt{2\log(2|\mathcal{W}|/\delta)}}{\sqrt{n}}$, the inequality becomes

$$\mathbb{P}\left(\left|\left(\mu_{\mathcal{D}} - \mu\right) \cdot w\right| \geq t\right) \leq \frac{\delta}{|\mathcal{W}|}$$

Use the Lipschitz condition and union bound of $w$, we have

$$\forall w \in \mathcal{W}, \quad \mathbb{P}\left(f\left(\mu_{\mathcal{D}} \cdot w\right) - f\left(\mu \cdot w\right) \geq L_f \cdot t\right) \leq \delta \tag{31}$$

Choose $w^* \in \mathcal{W}$ and combine with the inequality $f\left(\mu_{\mathcal{D}} \cdot w_{\mathcal{D}}\right) \leq f\left(\mu_{\mathcal{D}} \cdot w^*\right)$, we get the final result

$$f\left(\mu_{\mathcal{D}} \cdot w_{\mathcal{D}}\right) \leq f\left(\mu \cdot w^*\right) + \frac{\sqrt{2}L_f B_w}{\sqrt{n}}\sqrt{\log\frac{2|\mathcal{W}|}{\delta}} \tag{32}$$

with at least $1 - 3\delta$ probability. $\qquad\square$

Next, we prove term II that $f\left(\mu \cdot \overline{w}_K\right)$ is close to $f\left(\mu_{\mathcal{D}} \cdot \overline{w}_K\right)$.

**Lemma 6.** *We have*

$$f\left(\mu \cdot \overline{w}_K\right) \leq f\left(\mu_{\mathcal{D}} \cdot \overline{w}_K\right) + \frac{\sqrt{2}L_f B_w}{\sqrt{n}}\sqrt{\log\frac{2|\mathcal{W}|}{\delta}}$$

*with probability $1 - \delta$.*

*Proof.* This is easy to prove as we have a general result for all $w \in \mathcal{W}$ in equation (31). Here we take $\overline{w}_K \in \mathcal{W}$,

$$\mathbb{P}\left(f\left(\mu \cdot \overline{w}_K\right) - f\left(\mu_{\mathcal{D}} \cdot \overline{w}_K\right) \geq t\right) \leq \delta \tag{33}$$

which completes the proof. $\qquad\square$

Note that $\overline{w}_K$ violates the validity constraint in problem (5). This means that the calculated result may not satisfy the quality of the occupancy measure. To find the relation, we utilize the analogous lemmas in [Ozdaglar et al., 2023] to get the error bound between validity constraint violation and absolute objective difference.

**Lemma 7.** *Let $\overline{d}_K$ be a variable solved by Program (8) that violates the validity constraint (6) and $\overline{\pi}_K$ is the policy induced by $\overline{d}_K$. We can get that*

$$f(d_{\overline{\pi}_K}) \leq f(\overline{d}_K) + \frac{2B_w L_f \sqrt{2\log(|\mathcal{W}||\mathcal{X}|/\delta)}}{(1-\gamma)\sqrt{n}}$$

*with at least $1 - \delta$ probability.*

*Proof.* Define the marginalized occupancy measure as $\hat{d}_K(s) = \sum_{a \in \mathcal{A}} \overline{d}_K(s, a)$ and $\hat{d}_{\overline{\pi}_K}(s) = \sum_{a \in \mathcal{A}} d_{\overline{\pi}_K}(s, a)$. Then, we can write

$$\hat{d}_K(s) = \sum_{a \in \mathcal{A}} \overline{d}_K(s, a) \qquad \text{and} \qquad \hat{d}_{\overline{\pi}_K}(s) = \sum_{a \in \mathcal{A}} d_{\overline{\pi}_K}(s, a). \tag{34}$$

Let $P_\pi^{|\mathcal{S}||\mathcal{A}|} \in \mathbb{R}$ be the state transition matrix, i.e.,

$$P_{\pi_\mathcal{D}}(j, i) = \sum_{a \in A} P_{s^i, a}(s^j) \cdot \pi(a \mid s^i).$$

Also, we define the matrix $G_\pi = \text{Diag}(\pi_d(\cdot \mid s^1), \pi_d(\cdot \mid s^2), \cdots, \pi_d(\cdot \mid s^{|\mathcal{S}|})) \in \mathbb{R}^{|\mathcal{S}||\mathcal{A}| \times |\mathcal{S}|}$, and notice the fact that $MG_\pi = I - \gamma P_\pi$. Now, since $d_\pi$ satisfies the constraints in Problem, we have $Md_\pi = (1-\gamma)\mu_0$. This implies:

$$
\begin{aligned}
\|M\overline{d}_K - (1-\gamma)\mu_0\|_1 &= \|M(\overline{d}_K - d_{\overline{\pi}_K})\|_1 \\
&= \|MG_\pi(\hat{d}_K - \hat{d}_{\overline{\pi}_K})\|_1 \\
&= \|(I - \gamma P_\pi)(\hat{d}_K - \hat{d}_{\overline{\pi}_K})\|_1 \\
&\geq (1-\gamma)\|\hat{d}_K - \hat{d}_{\overline{\pi}_K}\|_1.
\end{aligned}
\tag{35}
$$

Here the last inequality is because $\gamma\|P_\pi(\hat{d}_K - \hat{d}_{\overline{\pi}_K})\|_1 \leq \gamma\|\hat{d}_K - \hat{d}_{\overline{\pi}_K}\|_1$, which follows from the fact that $P_\pi$ is a column stochastic matrix.

We have:

$$
|f(\overline{d}_K) - f(d_{\overline{\pi}_K})| \leq L_f|\overline{d}_K - d_{\overline{\pi}_K}| \tag{36}
$$
$$
= L_f|G_\pi(\hat{d}_K - \hat{d}_{\overline{\pi}_K})|
$$
$$
\leq L_f\|\hat{d}_K - \hat{d}_{\overline{\pi}_K}\|_1. \tag{37}
$$

The first inequality holds because of the Lipschitz condition of function $f$. So combining the above two inequalities, we get

$$
f(d_{\overline{\pi}_K}) - f(\overline{d}_K) \leq \frac{L_f\|M\overline{d}_K - (1-\gamma)\mu_0\|_1}{1-\gamma} \tag{38}
$$
$$
= \frac{L_f\|K\overline{w}_K - (1-\gamma)\mu_0\|_1}{1-\gamma} \tag{39}
$$

which follows the definition of density ratio $w$. Next, we give the bound of $\|K\overline{w}_K - (1-\gamma)\mu_0\|_1$.

From inequality (26) above we have that

$$\mathbb{P}\left(x^\top(K_\mathcal{D} - K)w \geq \zeta\right) \leq \delta, \quad \forall w \in \mathcal{W}, \ \forall x \in \mathcal{X}$$

Then take $w$ as $\overline{w}_K$ and $x$ as 1-norm, with at least $1 - \delta$ we have

$$
\begin{aligned}
\|K\overline{w}_K - (1-\gamma)\mu_0\|_1 &\leq \|K_\mathcal{D}\overline{w}_K - (1-\gamma)\mu_0\|_1 + \|(K - K_\mathcal{D})\overline{w}_K\|_1 \\
&\leq \zeta + \zeta
\end{aligned}
\tag{40}
$$

where the first inequality is because of the triangle inequality and the last inequality follows the fact that $\overline{w}_K$ is the optimal solution to problem (8) and the above inequality. Combining (40) and (38) completes the proof. $\qquad\square$

Now, we have the bound of term I and II. To further demonstrate the final bound, we also need to investigate the distance between the returned value $\overline{w}_K$ by primal-dual method and optimal solution $w_\mathcal{D}$.

Our algorithm 1 is the classical type of primal-dual subgradient method and our analysis mostly refers to [Nedić and Ozdaglar, 2009]. We put it here for the sake of completeness and add the theoretical analysis of our algorithm.

First, recall the Lagrange function and updated rules of the algorithm

$$
\begin{aligned}
\mathcal{L}(w, \lambda, \phi) &= f(\mu_\mathcal{D} \cdot w) + \lambda(\|K_\mathcal{D}w - (1-\gamma))\mu_0\|_1 - \zeta) + \phi(g(\mu_\mathcal{D} \cdot w) - \tau - \kappa) \\
w^{k+1} &= \mathcal{P}_\mathcal{W}\left[w^k - \eta\nabla\mathcal{L}_w(w^k, \lambda^k, \phi^k)\right] \\
\lambda^{k+1} &= \left[\lambda^k - \eta\nabla\mathcal{L}_\lambda(w^k, \lambda^k, \phi^k)\right]_0^{\lambda_{\max}^{k+1}} \\
\phi^{k+1} &= \left[\phi^k - \eta\nabla\mathcal{L}_\phi(w^k, \lambda^k, \phi^k)\right]_0^{\phi_{max}^{k+1}}
\end{aligned}
$$

For simplicity, we combine the Lagrange multipliers into one variable and call it $\nu$, accordingly, we set the constraint $g(w) \leq 0$. Note that constraints are analogous and we would split them in the end. Now the updated rules become

$$w^{k+1} = \mathcal{P}_{\mathcal{W}} \left[ w^k - \eta \nabla \mathcal{L}_w(w^k, \nu^k) \right] \tag{41}$$

$$\nu^{k+1} = \mathcal{P}_{\mathcal{M}} \left[ \nu^k - \eta \nabla \mathcal{L}_\nu(w^k, \nu^k) \right] \tag{42}$$

where $\nu^k \in \mathcal{M}$ and note that we turn the interval of $\nu$ into a projection set which is an equivalent transformation. The Lagrange function is denoted as

$$\mathcal{L}(w, \nu) = f(w) + \nu g(w) \tag{43}$$

**Theorem 2** (Saddle-Point Theorem in [Bertsekas, 1997]). *The pair $(w_\mathcal{D}, \nu_\mathcal{D})$ is a primal-dual optimal solution if and only if*

$$\mathcal{L}(w_\mathcal{D}, \nu) \leq \mathcal{L}(w_\mathcal{D}, \nu_\mathcal{D}) \leq \mathcal{L}(w, \nu_\mathcal{D}), \quad for \ all \ w \in \mathcal{W}, \ \nu \geq 0,$$

*where $w_\mathcal{D} \in \mathcal{W}$ and $\nu_\mathcal{D} \geq 0$.*

The analysis of convergence is based on this theorem and it is a standard result that characterizes the primal-dual optimal solutions as the saddle points of the Lagrange function[Nedić and Ozdaglar, 2009].

**Theorem 3.** *According to the updated rules, denote the solution to problem (8) as $\overline{w}_K$. Compared to the optimal solution $w_\mathcal{D}$, we have the upper bound for performance and constraint violation:*

$$(1) \quad \|g(\overline{w}_K)^+\| \leq \frac{2\|w_1 - w_\mathcal{D}\|}{K\eta} + \frac{2L}{\sqrt{K}} + \varepsilon \tag{44}$$

$$(2) \quad f(\overline{w}_K) \leq f^* + \frac{\|w_1 - w_\mathcal{D}\|^2}{2K\eta} + \eta L^2 \tag{45}$$

*where $\eta$ is the constant learning rate, $w_1$ is an initial state, $\varepsilon \geq 0$, $K$ is the iteration step and $L$ is the max subgradient of the Lagrange function. Furthermore, we denote, $w^* = w_\mathcal{D}$, $f(\overline{w}_K) = f(\mu_\mathcal{D} \cdot \overline{w}_K)$ and $f^* = f(\mu_\mathcal{D} \cdot w_\mathcal{D})$, we use the above notations for simplicity.*

We first introduce some lemmas and prove the theorem step by step.

**Lemma 8.** *Suppose that the sequence $\{w_k\}$ and $\{\nu_k\}$ are generated by the updated rules. Then we have*

$$(1) \quad \forall w \in \mathcal{W}, \quad \|w_{k+1} - w\|^2 \leq \|w_k - w\|^2 - 2\eta \left( \mathcal{L}(w_k, \nu_k) - \mathcal{L}(w, \nu_k) \right) + \eta^2 \|\mathcal{L}_w(w_k, \nu_k)\|^2.$$

$$(2) \quad \forall \nu \in \mathcal{M}, \quad \|\nu_{k+1} - \nu\|^2 \leq \|\nu_k - \nu\|^2 - 2\eta \left( \mathcal{L}(w_k, \nu_k) - \mathcal{L}(w_k, \nu) \right) + \eta^2 \|\mathcal{L}_\nu(w_k, \nu_k)\|^2.$$

*Proof.* We prove (1) and (2) is similar to (1).

$$\begin{aligned}
\|w_{k+1} - w\|^2 &= \|\mathcal{P}(w_k - \eta \mathcal{L}_w(w_k, \nu_k)) - w\|^2 \\
&\leq \|w_k - \eta \mathcal{L}_w(w_k, \nu_k) - w\|^2 \\
&= \|w_k - w\|^2 - 2\eta \mathcal{L}_w(w_k, \nu_k)(w_k - w) + \eta^2 \|\mathcal{L}_w(w_k, \nu_k)\|^2 \\
&\leq \|w_k - w\|^2 - 2\eta \left( \mathcal{L}(w_k, \nu_k) - \mathcal{L}(w, \nu_k) \right) + \eta^2 \|\mathcal{L}_w(w_k, \nu_k)\|^2
\end{aligned}$$

where the first equality comes from the update rule and the last inequality is because of the convexity of the Lagrange function. $\square$

**Assumption 6** (Boundness of subgradient). *We assume that the subgradient of the Lagrange function is bounded, such as*

$$\|\mathcal{L}_w(w_k, \nu_k)\| \leq L, \quad \|\mathcal{L}_\nu(w_k, \nu_k)\| \leq L, \quad \forall k \geq 0.$$

*where $L$ is the max Lipschitz constant in Lagrange function.*

**Lemma 9.** *Set $\overline{w}_k = \frac{1}{k}\sum_{i=1}^{k} w_{k_i}$ and $\overline{\nu}_k = \frac{1}{k}\sum_{i=1}^{k}\nu_{k_i}$, we have*

$$(1) \quad \frac{1}{k}\sum_{i=1}^{k}\mathcal{L}(w_i,\nu_i) - \mathcal{L}(w,\overline{\nu}_k) \le \frac{\|w_1 - w\|^2}{2\eta k} + \frac{\eta L^2}{2}, \quad \forall w \in \mathcal{W}.$$

$$(2) \quad -\frac{\|\nu_1 - \nu\|^2}{2\eta k} - \frac{\eta L^2}{2} \le \frac{1}{k}\sum_{i=1}^{k}\mathcal{L}(w_i,\nu_i) - \mathcal{L}(\overline{w}_k,\nu), \quad \forall \nu \in \mathcal{M}.$$

*Proof.* We only prove $(1)$ and equation $(2)$ is similar to $(1)$. By using Lemma 8 we have

$$\mathcal{L}(w_i,\nu_i) - \mathcal{L}(w,\nu_i) \le \frac{\|w_i - w\|^2 - \|w_{i+1} - w\|^2}{2\eta} + \frac{\eta L^2}{2}$$

Sum it from $1$ to $k$,

$$\frac{1}{k}\sum_{i=1}^{k}\mathcal{L}(w_i,\nu_i) - \mathcal{L}(w,\nu_i) \le \frac{\|w_1 - w\|^2 - \|w_k - w\|^2}{2\eta} + \frac{\eta L^2}{2}$$

$$\le \frac{\|w_1 - w\|^2}{2\eta} + \frac{\eta L^2}{2} \tag{46}$$

And combine with $\frac{1}{k}\sum_{i=1}^{k}\mathcal{L}(w,\nu_i) \ge \mathcal{L}(w,\overline{\nu}_k)$ which is because of Jensen's inequality, we have

$$\frac{1}{k}\sum_{i=1}^{k}\mathcal{L}(w_i,\nu_i) - \mathcal{L}(w,\overline{\nu}_k) \le \frac{\|w_1 - w\|^2}{2\eta} + \frac{\eta L^2}{2} \tag{47}$$

$\square$

**Assumption 7** (Slater's condition). *There exists a vector $\hat{w}$ such that*

$$g_j(\hat{w}) < 0, \quad \forall j = 1,2,\cdots,N$$

*we refer to $\hat{w}$ as a Slater's vector.*

Note that Slater's condition is assumed in many primal-dual method literature [Chen et al., 2022, Hong et al., 2024, Nedić and Ozdaglar, 2009].

**Lemma 10.** *Under Assumption 7, suppose $q^*$ is the dual optimal solution and $\overline{w}$ is a vector that satisfies Slater's condition, we have*

$$\|\nu\|_1 \le \frac{1}{\gamma}\left(f(\hat{w}) - q^*\right)$$

*where $\gamma = \min_{1 \le j \le m}\{-g_j(\hat{w})\}$ and $m$ is the number of constraints.*

The proof is referred to [Hiriart-Urruty and Lemaréchal, 1996]. The lemma above motivates the choice of dual set, that is

$$\mathcal{M} = \left\{\nu \ge 0 \mid \frac{f(\hat{w}) - \tilde{q}}{\gamma} + r\right\}$$

where $\tilde{q} \le q^*$ and $r \ge 0$ is a constant.

*Proof.* Now, we are ready to give the general proof of Theorem 3. Combine Lemma 8 and Assumption 6, we have

$$\|\nu_{k+1} - \nu\|^2 \le \|\nu_k - \nu\|^2 - 2\eta\left(\mathcal{L}(w_k,\nu_k) - \mathcal{L}(w_k,\nu)\right) + \eta^2\|\mathcal{L}_\nu(w_k,\nu_k)\|^2 \tag{48}$$

Moreover,

$$(\nu_i - \nu_\mathcal{D})\mathcal{L}_\nu(w_i,\nu_i) \le \mathcal{L}(w_i,\nu_i) - \mathcal{L}(w_i,\nu_\mathcal{D})$$
$$\le \mathcal{L}(w_i,\nu_i) - f^* \tag{49}$$

where the first equation is because of convexity of $\mathcal{L}_\nu$ and the last equation dues to the Slater's condition. We have, $\forall \nu \in \mathcal{M}$ and $i \geq 1$

$$
\begin{aligned}
(\nu - \nu_\mathcal{D})^\top \mathcal{L}_\nu (w_i, \nu_i) &= (\nu - \nu_\mathcal{D} + \nu_i - \nu_i)^\top \mathcal{L}_\nu (w_i, \nu_i) \\
&= (\nu - \nu_i)^\top \mathcal{L}_\nu (w_i, \nu_i) + (\nu_i - \nu_\mathcal{D}) \mathcal{L}_\nu (w_i, \nu_i) \\
&\leq \frac{\|\nu_i - \nu\|^2 - \|\nu_{i+1} - \nu\|^2}{2\eta} + \frac{\eta L^2}{2} + \mathcal{L} (w_i, \nu_i) - f^*
\end{aligned}
\tag{50}
$$

Then, sum over $i = 1, \cdots, k$,

$$
\sum_{i=1}^{k} (\nu - \nu_\mathcal{D})^\top \mathcal{L} (w_i, \nu_i) \leq \frac{\|\nu_1 - \nu\|^2}{2\eta} + \frac{\eta k L^2}{2} + \sum_{i=0}^{k-1} \mathcal{L} (w_i, \nu_i) - k f^*
\tag{51}
$$

Because the above function is for all $\nu$, then

$$
\max_{\nu \in \mathcal{M}} \left\{ \sum_{i=1}^{k} (\nu - \nu_\mathcal{D})^\top \mathcal{L} (w_i, \nu_i) \right\} \leq \frac{1}{2\eta} \max_{\nu \in \mathcal{M}} \|\nu_1 - \nu\|^2 \frac{\eta k L^2}{2} + \sum_{i=1}^{k} \mathcal{L} (w_i, \nu_i) - k f^*
\tag{52}
$$

Set

$$
s = \sum_{i=1}^{k} \mathcal{L}_\nu (w_i, \nu_i) = \sum_{i=1}^{k} g (w_i) \geq k g (\hat{w}_k)
\tag{53}
$$

Hence, if $s^+ = 0$, equation (52) holds. If $s^+ \neq 0$, define vector:

$$
\hat{\nu} = \nu_\mathcal{D} + r \frac{s^+}{\|s^+\|} \geq 0
$$

Combine Lemma 10, we have

$$
\|\hat{\nu}\| \leq \|\nu_\mathcal{D}\| + r \leq \frac{f (\hat{w}) - \tilde{q}}{\gamma} + r
\tag{54}
$$

which implies that for $\hat{\nu} \in \mathcal{M}$,

$$
\begin{aligned}
(\hat{\nu} - \nu_\mathcal{D})^\top s &= \sum_{i=1}^{k} (\hat{\nu} - \nu_\mathcal{D})^\top \mathcal{L} (w_i, \nu_i) \\
&\leq \max_{\nu \in \mathcal{M}} \left\{ \sum_{i=1}^{k} (\nu - \nu_\mathcal{D})^\top \mathcal{L} (w_i, \nu_i) \right\}
\end{aligned}
\tag{55}
$$

Recall the definition of $s$, we have

$$
\begin{aligned}
(\hat{\nu} - \nu_\mathcal{D})^\top s &= r \| \left[ \sum_{i=0}^{k} g (w_i) \right]^+ \| \\
&\leq r \max_{\nu \in \mathcal{M}} \left\{ \sum_{i=1}^{k} (\nu - \nu_\mathcal{D})^\top \mathcal{L} (w_i, \nu_i) \right\} \\
&\leq \frac{1}{2\eta} \max_{\nu \in \mathcal{M}} \|\nu_1 - \nu\|^2 + \frac{\eta k L^2}{2} + \sum_{i=1}^{k} \mathcal{L} (w_i, \nu_i) - k f^* \\
&\leq \frac{1}{2\eta} \max_{\nu \in \mathcal{M}} (\|\nu_1\| + \|\nu\|)^2 + \frac{\eta k L^2}{2} + \frac{\|w_1 - w_\mathcal{D}\|^2}{2\eta k} + \frac{\eta L^2}{2} \\
&\leq \frac{1}{2\eta} \max_{\nu \in \mathcal{M}} \|\nu\| + \frac{\eta k L^2}{2} + \frac{\|w_1 - w_\mathcal{D}\|^2}{2\eta k} + \frac{\eta L^2}{2} \\
&\leq \frac{1}{2\eta} \left[ \frac{f (\hat{w}) - \tilde{q}}{\gamma} + r \right] + \frac{\eta k L^2}{2} + \frac{\|w_1 - w_\mathcal{D}\|^2}{2\eta k} + \frac{\eta L^2}{2}
\end{aligned}
\tag{56}
$$

Then,

$$\|g(\overline{w}_k)^+\| \leq \frac{\|w_1 - w_{\mathcal{D}}\|^2}{2\eta k} + \frac{\eta L^2}{2} + \frac{2}{k\eta r}\left(\frac{f(\hat{w}) - \tilde{q}}{\gamma} + r\right)^2 \tag{57}$$

Until now, we assume the dual set is appropriately chosen in every iteration and we should quantify it. For the dual set

$$\mathcal{M} = \left\{\nu \geq 0 \big| \|\nu\| \leq \frac{f(\hat{w}) - \tilde{q}}{\gamma} + r\right\}$$

and the result in (57), We should choose

$$r = \min_{r \geq 0}\left\{\frac{\|w_1 - w_{\mathcal{D}}\|^2}{2\eta k} + \frac{\eta L^2}{2} + \frac{2}{k\eta r}\left(\frac{f(\hat{w}) - \tilde{q}}{\gamma} + r\right)^2\right\}$$

which is

$$r^*(k) = \sqrt{\left(\frac{f(\hat{w}) - \tilde{q}}{\gamma}\right)^2 + \frac{\|w_1 - w_{\mathcal{D}}\|^2}{4} + \frac{k\eta^2 L^2}{4}}, \quad \forall k \geq 1.$$

So the dual set in every iteration is

$$\mathcal{M}_k = \left\{\nu \geq 0 \big| \|\nu\| \leq \frac{f(\hat{w}) - \tilde{q}}{\gamma} + r^*(k)\right\}$$

which implies

$$\|g(\overline{w}_k)^+\| \leq \frac{4}{k\eta}\left(\frac{f(\hat{w}) - \tilde{q}}{\gamma} + \sqrt{\left(\frac{f(\hat{w}) - \tilde{q}}{\gamma}\right)^2 + \frac{\|w_1 - w_{\mathcal{D}}\|^2}{4} + \frac{k\eta^2 L^2}{4}}\right)$$

$$\leq \frac{4}{\eta k}\left[\frac{2(f(\hat{w}) - \tilde{q})}{\gamma} + \frac{\|w_1 - w_{\mathcal{D}}\|}{2} + \frac{\eta L \sqrt{k}}{2}\right] \tag{58}$$

For simplicity, we take the first term as a constant which doesn't affect our result:

$$\|g(\overline{w}_k)^+\| \leq \varepsilon + \frac{2\|w_1 - w_{\mathcal{D}}\|}{k\eta} + \frac{2L}{\sqrt{k}} \tag{59}$$

The result of (1) in Theorem 3 completes.

Next, we give the bound of the objective value. From the definition of the Lagrange function,

$$f(\overline{w}_k) \leq \frac{1}{k}\sum_{i=1}^{k} f(w_i) = \frac{1}{k}\sum_{i=0}^{k-1}\mathcal{L}(w_i, \nu_i) - \frac{1}{k}\sum_{i=1}^{k} g(w_i)\nu_i$$

Then,

$$f(\overline{w}_k) - f^* \leq \frac{1}{k}\sum_{i=1}^{k}\mathcal{L}(w_i, \nu_i) - \frac{1}{k}\sum_{i=1}^{k} g(w_i)\nu_i - f^*$$

$$\leq \frac{\|w_1 - w^*\|^2}{2\eta k} + \frac{\eta L^2}{2} - \frac{1}{k}\sum_{i=1}^{k}\nu_i g(w_i) \tag{60}$$

Note that

$$0 \in \mathcal{M} = \left\{\nu \geq 0 \big| \|\nu\| \leq \frac{f(\hat{w}) - \tilde{q}}{\gamma} + r\right\}$$

Set $\nu = 0$ and combine Lemma 8, we have

$$\|\nu_{k+1}\|^2 \leq \|\nu_k\|^2 + 2\eta\nu_k g(w_k) + \eta^2 L^2$$

Then
$$2\eta\nu_k g(w_k) \le \|\nu_k\|^2 - \|\nu_{k+1}\|^2 + \eta^2 L^2$$

Sum over $i = 0, \cdots, k - 1$,

$$-\sum_{i=1}^{k-1} \nu_i g(w_i) \le \frac{1}{2\eta}\left(\|w_1\|^2 - \|\nu_k\|^2\right) + \frac{\eta^2 L^2 k}{2} \tag{61}$$

We get the final result,

$$f(\overline{w}_k) \le f^* + \frac{\|\nu_1\|^2}{2\eta} + \frac{\|w_1 - w_{\mathcal{D}}\|^2}{2\eta k} + \frac{\eta^2 L^2}{2}$$

which completes the proof of Theorem 3. □

Then we have the all results to prove the regret bound in Theorem 1.

Theorem 3 indicates that after $K$ iterations the objective and violation bound for problem (8) are

$$f(\mu_{\mathcal{D}} \cdot \overline{w}_K) \le f(\mu_{\mathcal{D}} \cdot w_{\mathcal{D}}) + \frac{\|\nu_1\|^2}{2\eta} + \frac{\|w_1 - w_{\mathcal{D}}\|^2}{2\eta K} + \frac{\eta^2 L^2}{2}$$
$$= f(\mu_{\mathcal{D}} \cdot w_{\mathcal{D}}) + \frac{B^2}{2\eta K} + \frac{\eta^2 L^2}{2}$$
$$= f(\mu_{\mathcal{D}} \cdot w_{\mathcal{D}}) + \frac{B^2}{2\sqrt{K}} + \frac{L^2}{2K} \tag{62}$$

$$x^\top (K_{\mathcal{D}}\overline{w}_K - (1-\gamma)\mu_0) \le \zeta + \frac{2\|w_1 - w_{\mathcal{D}}\|}{K\eta} + \frac{2L}{\sqrt{K}} + \frac{\varepsilon}{\sqrt{K}}$$
$$= \zeta + \frac{2B}{\sqrt{K}} + \frac{2L}{\sqrt{K}} + \frac{\varepsilon}{\sqrt{K}} \tag{63}$$

$$g(h_{\mathcal{D}} \cdot \overline{w}_K) \le \tau + \kappa + \frac{2\|w_1 - w_{\mathcal{D}}\|}{K\eta} + \frac{2L}{\sqrt{K}} + \frac{\varepsilon}{\sqrt{K}}$$
$$= \tau + \kappa + \frac{2B}{\sqrt{K}} + \frac{2L}{\sqrt{K}} + \frac{\varepsilon}{\sqrt{K}} \tag{64}$$

where we set the initial Lagrange values equal to 0, the constant step size $\eta = \frac{1}{\sqrt{K}}$ and B is the distance between $w_1$ and optimal solution $w_{\mathcal{D}}$.

And recall Lemma 7, we have with at least $1 - 2\delta$ probability,

$$\|K\overline{w}_K - (1-\gamma)\mu_0\|_1 \le \|K_{\mathcal{D}}\overline{w}_K - (1-\gamma)\mu_0\|_1 + \|(K - K_{\mathcal{D}})\overline{w}_K\|_1$$
$$\le 2\zeta + \frac{\varepsilon}{\sqrt{K}} + \frac{2\|w_1 - w_{\mathcal{D}}\|}{K\eta} + \frac{2L}{\sqrt{K}} \tag{65}$$

Then, we have, with at least $1 - \delta$ probability, the bound is,

$$J_r(\overline{\pi}_K) - J_r(\pi^*)$$
$$= \underbrace{f(d_{\overline{\pi}_K}) - f(\overline{d}_K)}_{\text{I}} + \underbrace{f(\mu \cdot \overline{w}_K) - f(\mu_{\mathcal{D}} \cdot \overline{w}_K)}_{\text{II}} + \underbrace{f(\mu_{\mathcal{D}} \cdot \overline{w}_K) - f(\mu \cdot w^*)}_{\text{III}}$$
$$= \underbrace{f(d_{\overline{\pi}_K}) - f(\overline{d}_K)}_{\text{I}} + \underbrace{f(\mu \cdot \overline{w}_K) - f(\mu_{\mathcal{D}} \cdot \overline{w}_K)}_{\text{II}} + \underbrace{f(\mu \cdot \overline{w}_K) - f(\mu_{\mathcal{D}} \cdot w_{\mathcal{D}}) + f(\mu_{\mathcal{D}} \cdot w_{\mathcal{D}}) - f(\mu_{\mathcal{D}} \cdot w^*)}_{\text{III}}$$
$$\le \frac{L_f}{1-\gamma} \cdot \|K\overline{w}_K - (1-\gamma)\mu_0\|_1 + \frac{\sqrt{2}B_w L_f}{\sqrt{n}}\sqrt{\log\frac{2|\mathcal{W}|}{\delta}} + \frac{B^2}{2\sqrt{K}}$$
$$+ \frac{L^2}{2K} + \frac{\sqrt{2}B_w L_f}{\sqrt{n}}\sqrt{\log\frac{2|\mathcal{W}|}{\delta}}$$

$$\leq \frac{1}{1-\gamma}\left(\frac{6\sqrt{2}L_fB_w}{\sqrt{n}}\sqrt{\log\frac{2|\mathcal{W}||\mathcal{X}|}{\delta}} + \frac{2B}{\sqrt{K}} + \frac{2L}{\sqrt{K}} + \frac{B^2}{2\sqrt{K}} + \frac{L^2}{2K} + \frac{\varepsilon}{\sqrt{K}}\right)$$

$$\leq \frac{1}{1-\gamma}\left(\frac{6\sqrt{2}L_fB_w}{\sqrt{n}}\sqrt{\log\frac{2|\mathcal{W}||\mathcal{X}|}{\delta}} + \frac{\iota}{2\sqrt{K}}\right)$$

$$\leq \mathcal{O}(1/\sqrt{n}) + \mathcal{O}(1/\sqrt{K})$$

where we set $\iota = 4B + 4L + B^2 + \frac{L^2}{\sqrt{K}} + \varepsilon$.

**Proof of violation bound**

In the main text, we split the violation bound into three parts and list the relevant lemmas. Here we give the proofs for these lemmas and theorem.

**Proof of Lemma 1**

*Proof.* This proof is quite similar to Lemma 7 since we always want to establish connections between the required function and validity error bound. The main difference is the objective function and constraint function. So we keep the first half conclusion of the Lemma 7 which is

$$\|M\bar{d}_K - (1-\gamma)\mu_0\|_1 = \|M(\bar{d}_K - d_{\bar{\pi}_K})\|_1$$
$$\geq (1-\gamma)\|\hat{d}_K - \hat{d}_{\bar{\pi}_K}\|_1. \tag{66}$$

Then for the safety constraint function, we have

$$|g(\bar{d}_K) - g(d_{\bar{\pi}_K})| \leq L_g|\bar{d}_K - d_{\bar{\pi}_K}| \tag{67}$$
$$= L_g|G_\pi(\hat{d}_K - \hat{d}_{\bar{\pi}_K})|$$
$$\leq L_g\|\hat{d}_K - \hat{d}_{\bar{\pi}_K}\|_1. \tag{68}$$

where the first inequality is because Lipschitz condition of the safety constraint function. Also by combining the above functions, we have

$$g(\bar{d}_K) - g(d_{\bar{\pi}_K}) \leq \frac{L_g\|M\bar{d}_K - (1-\gamma)\mu_0\|_1}{1-\gamma} \tag{69}$$
$$= \frac{L_g\|K\overline{w}_K - (1-\gamma)\mu_0\|_1}{1-\gamma} \tag{70}$$

And in Lemma 7 we have already established the bound of $\|K\overline{w}_K - (1-\gamma)\mu_0\|_1$, then with at least $1 - 2\delta$ probability

$$g(\bar{d}_K) \leq g(d_{\bar{\pi}_K}) + \frac{2L_gB_w\sqrt{2\log(|\mathcal{W}||\mathcal{X}|/\delta)}}{(1-\gamma)\sqrt{n}}.$$

which completes the proof. $\square$

**Proof of Lemma 2**

*Proof.* In this lemma, actually, we will prove that $g(\mu \cdot \overline{w}_K)$ is close to $g(\mu_{\mathcal{D}} \cdot \overline{w}_K)$. We take $(\mu - \mu_{\mathcal{D}}) \cdot w$ as the random variable and it lies in $[-B_w, B_w]$. We have the Hoeffding inequality

$$\mathbb{P}\left(|\mu \cdot w - \mu_{\mathcal{D}} \cdot w| \geq t\right) \leq 2\exp\left(-\frac{2n^2t^2}{\sum_{i=1}^n(b_i - a_i)^2}\right) = \frac{\delta}{|\mathcal{W}|} \tag{71}$$

where we set $t = \frac{\sqrt{2}B_w}{\sqrt{n}}\sqrt{\log\frac{2|\mathcal{W}|}{\delta}}$. Next, take the union bound for $w$ and use the Lipschitz condition for $g$, we have

$$\forall w \in \mathcal{W}, \quad \mathbb{P}\left(g(\mu_{\mathcal{D}} \cdot w) - g(\mu \cdot w) \geq L_g \cdot t\right) \leq \delta \tag{72}$$

So for $\overline{w}_K \in \mathcal{W}$,

$$\mathbb{P}\left(g(\mu_{\mathcal{D}} \cdot \overline{w}_K) - g(\mu \cdot \overline{w}_K) \geq L_g \cdot t\right) \leq \delta \tag{73}$$

which means that with at least $1 - \delta$ probability,

$$g\left(\mu_{\mathcal{D}} \cdot \overline{w}_K\right) - g\left(\mu \cdot \overline{w}_K\right) \leq \frac{\sqrt{2}L_g B_w}{\sqrt{n}} \sqrt{\log \frac{2|\mathcal{W}|}{\delta}}.$$

The proof is complete. □

**Proof of Lemma 3**

*Proof.* This lemma is easy to be verified because we choose the relax parameter $\kappa$ is equal to $\frac{\sqrt{2}L_g B_w}{\sqrt{n}} \sqrt{\log \frac{2|\mathcal{W}|}{\delta}}$ and recall the result in Theorem 3 which completes the proof. □

Thus, take the Lemma 1, 2 and 3 together, we have, with at least $1 - 3\delta$,

$$J_c(\overline{\pi}_K) - \tau = \underbrace{g(d_{\overline{\pi}_K}) - g(\overline{d}_K)}_{\text{I}} + \underbrace{g(\mu \cdot \overline{w}_K) - g(\mu_{\mathcal{D}} \cdot \overline{w}_K)}_{\text{II}} + \underbrace{g(\mu_{\mathcal{D}} \cdot w_{\mathcal{D}}) - \tau}_{\text{III}}$$

$$\leq \frac{L_g}{1-\gamma} \cdot \|K\overline{w}_K - (1-\gamma)\mu_0\|_1 + \frac{\sqrt{2}B_w}{\sqrt{n}} \sqrt{\log \frac{2|\mathcal{W}|}{\delta}} + \kappa + \frac{2B}{\sqrt{K}} + \frac{2L}{\sqrt{K}} + \frac{\varepsilon}{\sqrt{K}}$$

$$\leq \frac{1}{1-\gamma} \left( \frac{6\sqrt{2}L_g B_w}{\sqrt{n}} \sqrt{\log \frac{2|\mathcal{W}||\mathcal{X}|}{\delta}} + \frac{2B}{\sqrt{K}} + \frac{2L}{\sqrt{K}} + \frac{\varepsilon}{\sqrt{K}} \right) + \frac{2B}{\sqrt{K}} + \frac{2L}{\sqrt{K}} + \frac{\varepsilon}{\sqrt{K}}$$

$$\leq \frac{6\sqrt{2}L_g B_w}{(1-\gamma)\sqrt{n}} \sqrt{\log \frac{2|\mathcal{W}||\mathcal{X}|}{\delta}} + \frac{\upsilon}{\sqrt{K}}$$

$$\leq \mathcal{O}(1/\sqrt{n}) + \mathcal{O}(1/\sqrt{K})$$

where we set $\upsilon = \frac{1}{1-\gamma}(4B + 4L + 2\varepsilon)$ which completes the proof.

## B Experiments

We introduce a practical version of our primal-dual algorithm, following the key structure in Algorithm 1. We utilize this algorithm to conduct various experiments in this paper.

---

**Algorithm 2:** Practical version of POCC

1 **Input**: Dataset $D = \{(s_i, a_i, r_i, c_i)\}_{i=1}^n$, cost threshold $\tau$, learning rate $\eta_\lambda, \eta_w, \eta_\phi$, relaxed parameter $\zeta, \kappa$.;
2 Initialize network of importance weight $w_\psi$;
3 **for** $k = 1, 2, \ldots, K$ **do**
4      Jointly optimize:

$$w_\psi^{k+1} = \text{Adam}\left(w_\psi^k - \eta_\phi \nabla L(w_\psi, \lambda^k, \phi^k)\right),$$
$$\lambda^{k+1} = \text{Adam}\left(\lambda^k - \eta_\lambda \nabla L(w_\psi^k, \lambda, \phi^k)\right),$$
$$\phi^{k+1} = \text{Adam}\left(\phi^k - \eta_\tau \nabla L(w_\psi^k, \lambda^k, \phi)\right).$$

5 Compute the importance weight: $\forall (s, a), \;\; w(s, a) = w_\psi(s, a)$;
6 Extract the policy: $\pi(a \mid s) = \frac{w_\psi(s,a)\pi_\mu(a|s)}{\sum_{a' \in A} w_\psi(s,a')\pi_\mu(a'|s)}$;
7 **Output:** Policy $\pi$ ;

---

**Hyperparameters** In the experiments of maze and FrozenLake environments, we set the discount factor as $\gamma = 0.99$ and the cost threshold as $\tau = 0$, indicating that the agent is not supposed to incur any cost. For our algorithm 2 and COptiDICE [Lee et al., 2021], we perform a grid search on the set $\{0.00001, 0.00005, 0.0001, 0.0005, 0.001, 0.005\}$ to determine the learning rate for parameters. We ultimately select a learning rate of 0.00001 for the neural network and a Lagrange scalar learning

rate of 0.0001. We set $\zeta = 0.1$ and $\kappa = 0.001$ in our algorithm, and we set $\alpha = 0.5$ in COptiDICE. In both environments, we run the algorithm for $K = 100000$ iterations to ensure model convergence. Additionally, we use fully-connected neural networks with a single hidden layer of width 64.

**Continuous environments**   We also run a list of experiments in SafetyGym with offline datasets provided by [Liu et al., 2023a] and compare with comprehensive baselines (e.g., CPQ in [R5], PDCA in [R4], CoptiDICE in [R6], and BEAR-Lagrangian in [R7], [R8]). Note that only PDCA and our algorithm provide theoretical results. In these experiments, to deal with the continuous state-action space, we use the fully connected single hidden-layer neural network of width 128 to represent $w$. We summarize the evaluation results in the following table. All the rewards and costs are normalized and the cost threshold is 1. Each value is averaged over 20 evaluation episodes and 3 random seeds.

Table 2: All the rewards and costs are normalized. The cost threshold is 1. Blue: Safe agents with the highest reward.

| Task | COptiDICE [Reward, Cost] | CPQ [Reward, Cost] | BEAR-Lag [Reward, Cost] | PDCA [Reward, Cost] | Ours [Reward, Cost] |
|---|---|---|---|---|---|
| AntRun | [0.6, 0.94] | [0.03, 0.02] | [0.15, 0.73] | [0.28, 0.93] | [0.6, 0.01] |
| CarRun | [0.87, 0.0] | [0.95, 1.79] | [0.68, 7.78] | [0.91, 0.0] | [0.90, 0.0] |
| BallRun | [0.59, 3.52] | [0.22, 1.27] | [−0.47, 5.03] | [0.55, 3.38] | [0.24, 0.0] |
| BallCircle | [0.70, 2.61] | [0.64, 0.76] | [0.86, 3.09] | [0.63, 2.29] | [0.39, 0.93] |
| CarPush1 | [0.23, 0.5] | [−0.03, 0.95] | [0.21, 0.54] | [0.17, 0.41] | [0.20, 0.4] |

**Compute setting**   We run the experiments with NVIDIA GeForce RTX 3080 Ti 8-Core Processor.

