# OpenReview forum: "Safe and Efficient: A Primal-Dual Method for Offline Convex CMDPs under Partial Data Coverage"
_NeurIPS.cc/2024/Conference — NeurIPS 2024 poster_

### Official Review · Reviewer_H1LM · 2024-06-13

**Soundness:** 3
**Presentation:** 3
**Contribution:** 3
**Rating:** 6
**Confidence:** 4

**Summary:**

This paper formulates offline convex MDPs under safety constraints and proposes a linear programming-based primal-dual algorithm for solving it. The authors make a partial data coverage assumption yet achieve a sample complexity of $\mathcal{O}(\frac{1}{(1-\gamma)\sqrt{n}})$ while the current SOTA is $\mathcal{O}(\frac{1}{(1-\gamma)^2\sqrt{n}})$. This paper also conducted the empirical evaluation and showed that the proposed algorithm achieves better performance in terms of reward and safety.

**Strengths:**

- Offline convex CMDPs are promising formulation that covers wide-ranging problems including safe imitation learning or standard offline CMDPs.
- The theoretical results are strong. The sample complexity is improved by $1-\gamma$ compared to the SOTA, which is an important contribution to the offline safe RL community. The assumptions the authors make are 1) concentrability of the optimal policy, 2) realizability, 3) completeness, 4) Boundness of $\mathcal{W}$ and $\mathcal{X}$, and 5) Lipschitz continuity of the $f$ and $g$. Though the number of assumptions is large and each assumption is strong, all of them are standard assumptions in previous work and we understand that it is almost impossible to prove sample complexity without them.
- While this paper is highly theoretical, the authors also provide empirical results. I think it is nice to evaluate their algorithm in two settings: safe imitation learning and offline safe RL.

**Weaknesses:**

- While I admit that the main contribution of this paper is theory, I consider that the empirical experiments are not fully conducted. Both environments are toy problems, and there is no baseline method in 5.1. In 5.2, the benchmark task is the frozen lake, which is a much easier task than previous work tried to solve. I know COptiDICE is a well-known baseline algorithm, but there are many more algorithms that perform better than COptiDICE.
- The authors say "We include the limitations in our assumptions", but I do not think that the limitations of this paper exist except for the assumptions. I would recommend that the authors discuss the limitations such as scalability, empirical performance in more complicated tasks, etc.

**Minor Comments and Typos**
- Line 45: The following sentence seems weird to me. Should this be "We formulate ..."?
    - "We for the first study the offline."
- Figure 3: (a), (b), ..., (d) do not exist in Figure 3 while they are mentioned in the caption.

**Questions:**

- In lines 229 - 238, the authors claim as follows. However, the empirical results are provided for grid-world environments. The size of the grid world is rather small (i.e., $8 \times 8$). Could you tell me why the authors used such small environments? I personally think that the following claim is not fully supported.
> Besides, our algorithm is appropriate in the scenario with large-scale state-action space
- I think I could follow the theorem and proof, but could you tell me what enables the authors to achieve the SOTA sample complexity $\mathcal{O}(\frac{1}{(1-\gamma)\sqrt{n}})$? What is the biggest reason why the author could improve the current SOTA by a factor of $1/(1-\gamma)$?

**Limitations:**

Though the authors discuss the limitations regarding assumptions, I think there are other limitations such as the applicability to more complicated tasks with continuous state-action spaces, computational complexity, etc.

---

> ### Author Rebuttal · Authors · 2024-08-07
>
> Thank you for your positive feedback on our paper! We appreciate your support and comments. We'd like to respond to the major comments in the following.
>
> **The experiments are toy examples and have limitations when extending to complicated continuous state-action space.**: As our paper is primarily on the theory side, the experiments are used to justify the theoretical results, including the data coverage, function approximation assumption, and the sample complexity of our algorithm. According to the comprehensive experiments in [R1] and [R2], CoptiDICE is the most stable and effective algorithm, so we consider CoptiDICE as the baseline in our submission.
>
> But, following the reviewer's suggestions, we conduct a list of experiments in the challenging and continuous environment--SafetyGym with comprehensive baselines (e.g., CPQ in [R3], PDCA in [R1], CoptiDICE in [R2], and BEAR-Lagrangian in [R4], [R5]). Note that only PDCA and our algorithm provide theoretical results. In these experiments, to deal with the continuous state-action space, we use the fully connected single hidden-layer neural network of width 128 to represent $w$.
>
> We summarize the evaluation results in the following table. All the rewards and costs are normalized. The cost threshold is 1. Each value is averaged over 20 evaluation episodes and 3 random seeds.
>
> |**Task**|**COptiDICE**||**CPQ**||**BEAR-Lag**||**PDCA**||**Ours**||
> |-|-|-|-|-|-|-|-|-|-|-|
> ||**[Reward,Cost]**||**[Reward,Cost]**||**[Reward,Cost]**||**[Reward,Cost]**||**[Reward,Cost]**||
> |**AntRun**|**[0.6,0.94]**|| [0.03,0.02]||[0.15,0.73]||[0.28,0.93]||**[0.6,0.01]**||
> |**BallRun**|[0.59, 3.52]||[0.22,1.27]||[-0.47,5.03]||[0.55,3.38]||**[0.24,0.0]**||
> |**BallCircle**|[0.70,2.61]||**[0.64,0.76]**||[0.86,3.09]||[0.63,2.29]||[0.39,0.93]||
> |**CarPush1**|[0.23,0.5]||[-0.03,0.95]||**[0.21,0.54]**||[0.17,0.41]||[0.20,0.4]||
> |**CarRun**|[0.87,0.0]||[0.95,1.79]||[0.68,7.78]||[0.85,0.0] ||**[0.90,0.0]**||
>
> The results show our algorithm is the only one to guarantee safety across all environments. In AntRun, BallRun, and CarRun tasks, our algorithm can achieve the highest reward and small cost among all baseline algorithms.
> Note our algorithm is very flexible to incorporate more sophisticated function approximations (e.g., deeper and more advanced neural networks) to potentially achieve better empirical performance.
>
> **Question about SOTA sample complexity**: We think the most important reason that we can achieve the SOTA sample complexity is that our algorithm is direct enough. Our algorithm actually is to measure the distance between program (5)-(7) and the approximate program (8)-(10). In our proof, the discounted factor $\gamma$ only exists in Lemma 2 which depicts the distance between the reward and the constraint violation. However, most previous works are more complex compared with ours. For example, the paper [R1] has to use two oracles in their algorithm, introducing additional factors $(1-\gamma)^{-1}$, which makes their result weaker than ours.
>
> **Minor comments and Typos**: Thank you for your detailed read and we will fix it in our paper.

---

> ### Author Response · Authors · 2024-08-07
> **References**
>
> [R1]: Kihyuk Hong, Yuhang Li, and Ambuj Tewari. "A primal-dual-critic algorithm for offline constrained
> reinforcement learning." In International Conference on Artificial Intelligence and Statistics, pages
> 280–288. PMLR, 2024.
>
> [R2]: Jongmin Lee, Cosmin Paduraru, Daniel J Mankowitz, Nicolas Heess, Doina Precup, Kee-Eung Kim,
> and Arthur Guez. "Coptidice: Offline constrained reinforcement learning via stationary distribution
> correction estimation." In International Conference on Learning Representations, 2021.
>
> [R3]: Haoran Xu, Xianyuan Zhan, and Xiangyu Zhu. "Constraints penalized q-learning for safe offline
> reinforcement learning." In Proceedings of the AAAI Conference on Artificial Intelligence, volume 36,
> pages 8753–8760, 2022.
>
> [R4]: Aviral Kumar, Justin Fu, Matthew Soh, George Tucker, and Sergey Levine. "Stabilizing off-policy
> q-learning via bootstrapping error reduction." Advances in neural information processing systems,
> 32, 2019.
>
> [R5]: Adam Stooke, Joshua Achiam, and Pieter Abbeel. "Responsive safety in reinforcement learning
> by pid lagrangian methods." In International Conference on Machine Learning, pages 9133–9143.
> PMLR, 2020.

---

> > ### Comment · Reviewer_H1LM · 2024-08-08
> > **Responses**
> >
> > Thank you for the clarification and additional experiments. The concerns I had at the time of the initial review are resolved. After reading other reviews and authors' rebuttals, I still recommend acceptance of this paper. That said, I think that the impact of this paper would be moderate-to-high, so I keep the original score of 6.
> >
> > > 6: Weak Accept: Technically solid, moderate-to-high impact paper, with no major concerns with respect to evaluation, resources, reproducibility, ethical considerations.

---

> > > ### Author Response · Authors · 2024-08-08
> > >
> > > Thanks a lot for your precious time in helping us improve our paper. Much appreciated! Please let us know if you have any follow-up questions. We will be happy to answer them.

---

### Official Review · Reviewer_9gP2 · 2024-06-29

**Soundness:** 2
**Presentation:** 2
**Contribution:** 3
**Rating:** 5
**Confidence:** 4

**Summary:**

This paper investigates batch RL with safety constraints and function approximation, which is a question of both theoretical and practical importance.

**Strengths:**

The assumptions considered in this paper are less restrictive compared to previous works. In particular, most previous works consider linear objective, and this paper relaxes linearity to convexity, which makes the results apply to a broader class of learning problems (e.g. batch RL with entropy regularization). It is also good to see the experimental results.

**Weaknesses:**

(1) The presentation is a bit messy. The class X is not well-explained. The Lipschitz constant L appeared in Theorem 1 is not defined in the main text. Is L bounded by L_f, L_g and other parameters?

(2) There is an unstated assumption: the convergence of the proposed algorithm actually relies on the convexity of W (Lemma 8).

(3) The upper bounds are stated in terms of log|W|, the log-cardinality of the function class W. However, it may be problematic to assume W is finite: if W is both convex and finite, then W must be a singleton, and the results are vacuous. Therefore, the upper bounds should instead be stated in terms of the log covering number of W.

**Questions:**

Q1. Is there a relationship between the proposed primal-dual update rules and the actor-critic algorithms?

Q2. In assumption 4, it is required that all functions in W have a uniform upper bound B_w, which implies $B_w\geq C_\pi^\star $. It would be better to explicitly state that it requires prior knowledge on the concentrability constant.

Q3. The class X does not appear in the algorithm. Is it introduced purely for the sake of analysis?

---

> ### Author Rebuttal · Authors · 2024-08-07
>
> Thank you for your insightful comments. We would like to address your concerns point by point.
>
> **The class $\mathcal X$ is not well-explained**: In Assumption 3, operator $\phi(w)$ or $x$ aims to calculate the $l_1$ norm of the constraint function $Kw - (1-\gamma)\mu_0$. It assumes that for all $w \in \mathcal W$, $l_1$ norm of the constraint function exists and thus can be used to calculate the validity constraint violation (or called Bellman error). Only when the constraint violation is a computable quantity can we relax the equality constraint (6) and obtain an approximate program (8)-(10). So class $\mathcal X$ is used to depict the validity constraint violation and simplify our analysis. We will add this explanation to our paper.
>
> **The convergence of the proposed algorithm relies on the convexity of $\mathcal W$**: Thank you for your detailed discussion and we will add this to our revision. In fact, if we want program (8)-(10) to be computationally tractable, function class $\mathcal W$ is supposed to be convex. However, when $\mathcal W$ is not convex, the operation of convexification on function class $\mathcal W$ is reasonable and common in offline safe RL [R1, R2]. Even if $w$ is parameterized by the neural network, as studied in [R3], when the neural network is over-parameterized, it would have some almost convex properties such that stochastic gradient descent (SGD) can find global minima on the training objective of neural networks. Thus in our algorithm, if the neural network is over-parameterized, it will also have the property of convergence.
>
> **Question about the log-cardinality of the function class $\mathcal W$**: In our paper, we do not assume the function class $\mathcal W$ is finite. When $\mathcal W$ is a continuous set, the cardinality represents the covering number or the number of extreme points of the function class [R4], and it does not affect our results.
>
> **Relationship between the proposed primal-dual update rules and the actor-critic algorithm**: We think that our algorithm can be viewed from the perspective of actor-critic in some sense. For example, as studied in [R5, R6, R7], in the LP formulation of CMDP, the Lagrange multipliers of Bellman flow constraints are value functions. Because the value function is the dual variable of the occupancy measure in LP formulation. So the Lagrange multiplier $\lambda$ in our algorithm refers to the critic in actor-critic algorithm. And $w$ refers to the actor in actor-critic algorithm since $w$ equals to the occupancy measure in some sense and represents the policy in our algorithm.
>
> **The Lipschitiz constant $L$ and the prior knowledge on concentrability constant**: The Lipschitz constant $L$ is defined in the appendix and it is the upper bound of $L_f$ and $L_g$. And $B_w$ actually requires prior knowledge on the concentrability constant. Thank you for your detailed advice and we will add these to our paper.

---

> ### Author Response · Authors · 2024-08-07
> **References**
>
> [R1]: Hoang Le, Cameron Voloshin, and Yisong Yue. "Batch policy learning under constraints." In International Conference on Machine Learning, pages 3703–3712. PMLR, 2019.
>
> [R2]: Kihyuk Hong, Yuhang Li, and Ambuj Tewari. "A primal-dual-critic algorithm for offline constrained reinforcement learning." In International Conference on Artificial Intelligence and Statistics, pages 280–288. PMLR, 2024.
>
> [R3]: Zeyuan Allen-Zhu, Yuanzhi Li, and Zhao Song. "A convergence theory for deep learning via over-parameterization. In International conference on machine learning." pages 242–252. PMLR, 2019.
>
> [R4]: Asuman E Ozdaglar, Sarath Pattathil, Jiawei Zhang, and Kaiqing Zhang. "Revisiting the linear-
> programming framework for offline rl with general function approximation." In International
> Conference on Machine Learning, pages 26769–26791. PMLR, 2023.
>
> [R5]: Ofir Nachum, Yinlam Chow, Bo Dai, and Lihong Li. "Dualdice: Behavior-agnostic estimation of
> discounted stationary distribution corrections." Advances in neural information processing systems,
> 32, 2019.
>
> [R6]: Jongmin Lee, Wonseok Jeon, Byungjun Lee, Joelle Pineau, and Kee-Eung Kim. "Optidice: Offline
> policy optimization via stationary distribution correction estimation." In International Conference
> on Machine Learning, pages 6120–6130. PMLR, 2021.
>
> [R7]: Jongmin Lee, Cosmin Paduraru, Daniel J Mankowitz, Nicolas Heess, Doina Precup, Kee-Eung Kim, and Arthur Guez. "Coptidice: Offline constrained reinforcement learning via stationary distribution correction estimation." In International Conference on Learning Representations, 2021.

---

> ### Comment · Reviewer_9gP2 · 2024-08-08
>
> Thank you for your response. Some comments:
>
> > However, when W is not convex, the operation of convexification on function class W is reasonable and common in offline safe RL
>
> While the algorithm can operate on the convexified function class, the log covering number of the convexified function class can be much larger than the log covering number of W. Therefore, when you compare your results with existing works, it is necessary to highlight the necessity of convexity.
>
> > In our paper, we do not assume the function class W is finite
>
> I don't think this is an honest claim, given that your current proof of Lemma 5 directly uses the union bound over W. You don't even mention how cardinality is defined for continuous W: The word "covering" does not appear in the paper. Further, if it is indeed defined as the covering number, it has to depend on the covering radius, which is also not mentioned.
>
> Of course, this is a relatively minor issue. However, you shall admit that it is a mistake, instead of denying it.

---

> > ### Author Response · Authors · 2024-08-09
> >
> > We greatly appreciate your comments on the function class $\mathcal W$ and would like to respond to them as follows.
> >
> > **When comparing our results with existing works, it is necessary to highlight the necessity of convexity:** We will highlight our results based on the convex or convexified function class $\mathcal W$ and clarify that this is consistent with our most related work [7, 14, 17] (in Table 1) as they also require a similar convex property (either a tabular setting in [7] or convexified policy class in [14, 17]).
> >
> > **Elaboration on the function class $\mathcal W$:** We apologize for any misunderstandings and for any unintentional impression of dishonesty regarding our claim on the function class $\mathcal W$ due to the missing definition and statement. We would certainly clarify the definition of $\mathcal W$ and $|\mathcal W|$ (e.g., emphasize the continuous property and the covering radius and number).
> >
> > We thank the reviewer once again for the detailed comments, which have definitely helped improve the quality of our paper. We sincerely hope our response addresses your major concerns and that you will consider reevaluating our work. Please let us know if you have any further comments, and we will do our best to address them.

---

> > > ### Comment · Reviewer_9gP2 · 2024-08-11
> > >
> > > Given your responses and the promised changes, I will raise my rating to 5, favoring the acceptance of this paper.

---

> > > > ### Author Response · Authors · 2024-08-11
> > > >
> > > > We sincerely thank you for the constructive suggestions that helps us improve our paper. Much appreciated! Please let us know if you have any follow-up questions. We will be happy to answer them.

---

### Official Review · Reviewer_Za97 · 2024-07-11

**Soundness:** 3
**Presentation:** 2
**Contribution:** 3
**Rating:** 6
**Confidence:** 3

**Summary:**

This paper proposes a novel linear programming based primal-dual algorithm for convex MDPs which incorporates “uncertainty” parameters to improve data efficiency, while requiring only partial data coverage assumption. The authors provide theoretical results achieve a sample complexity of $O(1/((1-\gamma)\sqrt{n}))$ under general function approximation, improving the current state-of-the-art by a factor of $1/(1 − \gamma)$, where $n$ is the number of data samples in an offline dataset, and $\gamma$ is the discount factor. The authors also run experiments to validate their theoretical findings, which demonstrate the practical efficacy of their approach in achieving improved safety and learning efficiency in safe offline settings.

**Strengths:**

1.	The studied problem, i.e., safe offline RL in convex MDPs, is well-motivated and can be applied to autonomous driving, robotics, etc.
2.	The authors design a novel linear programming based primal-dual algorithm for convex MDPs under only the partial coverage assumption, instead of the full coverage assumption. The authors also provide sample complexity for this algorithm, which improves the current result by a factor of $1/(1 − \gamma)$.
3.	Empirical evaluations are also presented to validate the practical efficacy of the proposed algorithm.

**Weaknesses:**

1.	Can the authors give more explanations on Assumptions 2-4. Why do you introduce the set $\mathcal{W}$ and $\mathcal{X}$. Does the algorithm need to know these two sets in advance? If yes, what does it mean in practice?
2.	Does the algorithm need to know the behavior policy $\mu(a|s)$ in advance (in Eq. (12))? This is not a practical assumption.
3.	The authors should give more comparison to the existing results for offline (linear) constrained RL, since (linear) constrained MDPs is an important example of convex MDPs.
4.	What are the technical challenges and novelty in the offline convex MDP (RL) problem, compared to the existing online convex MDP (RL) works?

**Questions:**

Please see the weaknesses above.

**Limitations:**

Please see the weaknesses above.

---

> ### Author Rebuttal · Authors · 2024-08-07
>
> Thank you for your positive feedback and detailed comments on our paper. We will respond to the major concern in the following.
>
> **Need more explanations on Assumptions 2-4 and the reason for introducing the set $\mathcal W$ and $\mathcal X$.**: Assumption 2 assumes the optimal policy $\pi^*$ $(w^*)$ is included in the function class $\mathcal W.$ It means (5)--(7) (with $w \in \mathcal W$) is a proper baseline as it includes the optimal policy. Assumption 3 is a completeness-type assumption. Operator $\phi(w)$ or $x$ aims to calculate the $l_1$ norm of the constraint function $Kw - (1-\gamma)\mu_0$. It assumes that for all $w \in \mathcal W$, $l_1$ norm of the constraint function exists and thus can be used to calculate the constraint violation (or called Bellman error). Only when the constraint violation is a computable quantity can we relax the equality constraint (6) and obtain an approximate program (8)-(10). Assumption 4 is a standard assumption in offline RL that assumes the boundness of function classes $\mathcal W$ since we always do not want variable $w$ to be infinite.
>
> The introduction of set $\mathcal W$ and $\mathcal X$ helps our analysis. In our algorithm, we want to measure the distance between program (5)-(7) and the approximate program (8)-(10). So if we assume the optimal solutions to program (5)-(7) and program (8)-(10) are in the same function class $\mathcal W$, the distance between the above two programs can be directly in function class $\mathcal W$. Moreover, $x \in \mathcal X$ is used to calculate the $l_1$ norm of the constraint function and further measure the constraint violation as we state above. Actually, we do not need to know set $\mathcal W$ and $\mathcal X$ in advance since they are used for the sake of analysis.
>
> Thank you for your advice, we will further explain this in our paper.
>
> **Does the algorithm need to know the behavior policy in advance?**: The algorithm does not need to know the behavior policy $\mu(a | s)$ in advance. For ease of exposition, we assume the behavior policy is known. When it is unknown in practice, behavior clone is an effective approach to extract the behavior policy from the dataset. Specifically, we can estimate the learned behavior policy $\hat{\pi}$ by $\hat{\pi} (a | s) = \frac{n(s,a)}{n(s)}$, where $n(s,a)$ is the number of $(s,a)$ state-action pairs in the offline dataset. It can be shown that the gap between the learned policy $\hat{\pi}$ and the real behavior policy $\pi_\mu$ is upper bounded by $\min ( 1, |\mathcal S| / n )$ [R1], which does not affect our sample complexity. Furthermore, the experiments in our paper do not assume the behavior policy and utilize the behavior clone to estimate the behavior policy, where our algorithms also achieve great results.
>
> **More comparison to the existing results for offline (linear) constrained RL**: In theory, our algorithm achieves state-of-the-art results with general function approximation under partial data coverage. However, in previous works, [R2] proposes a meta-algorithm combining with general function approximation but achieves $\mathcal O \left( \frac{1}{(1-\gamma)^5 \sqrt{n}}\right)$ sample complexity under the strong Bellman completeness assumption. [R3] also focuses on occupancy measures but their algorithm can just apply to discrete state-action spaces. The most related work [R4] approaches the problem from the perspective of the Actor-Critic algorithm, analyzing the sample complexity to be $\mathcal O \left( \frac{1}{(1-\gamma)^2 \sqrt{n}}\right)$ under a stronger full data coverage assumption.
>
> In experiments, following the reviewer's suggestions, we conduct a list of experiments in the challenging and continuous environment--SafetyGym with comprehensive baselines (e.g., CPQ in [R5], PDCA in [R4], CoptiDICE in [R6], and BEAR-Lagrangian in [R7], [R8]). Note that only PDCA and our algorithm provide theoretical results. In these experiments, to deal with the continuous state-action space, we use the fully connected single hidden-layer neural network of width 128 to represent $w$.
>
> We summarize the evaluation results in the following table. All the rewards and costs are normalized. The cost threshold is 1. Each value is averaged over 20 evaluation episodes and 3 random seeds.
>
> |**Task**|**COptiDICE**||**CPQ**||**BEAR-Lag**||**PDCA**||**Ours**||
> |-|-|-|-|-|-|-|-|-|-|-|
> ||**[Reward,Cost]**||**[Reward,Cost]**||**[Reward,Cost]**||**[Reward,Cost]**||**[Reward,Cost]**||
> |**AntRun**|**[0.6,0.94]**|| [0.03,0.02]||[0.15,0.73]||[0.28,0.93]||**[0.6,0.01]**||
> |**BallRun**|[0.59, 3.52]||[0.22,1.27]||[-0.47,5.03]||[0.55,3.38]||**[0.24,0.0]**||
> |**BallCircle**|[0.70,2.61]||**[0.64,0.76]**||[0.86,3.09]||[0.63,2.29]||[0.39,0.93]||
> |**CarPush1**|[0.23,0.5]||[-0.03,0.95]||**[0.21,0.54]**||[0.17,0.41]||[0.20,0.4]||
> |**CarRun**|[0.87,0.0]||[0.95,1.79]||[0.68,7.78]||[0.85,0.0] ||**[0.90,0.0]**||
>
> The results show our algorithm is the only one to guarantee safety across all environments. In AntRun, BallRun, and CarRun tasks, our algorithm can achieve the highest reward and small cost among all baseline algorithms.
> Note our algorithm is very flexible to incorporate more sophisticated function approximations (e.g., deeper and more advanced neural networks) to potentially achieve better empirical performance.
>
> **The challenge between online convex MDP and offline convex MDP.**: We think the main challenge between online convex MDPs and offline convex MDPs lies in data, including data quality and data assumptions. For example, in R[9], they propose a method that uses standard RL algorithms to solve convex MDPs. However, in the offline setting, each RL algorithm has its own data assumption. It is challenging to satisfy all data assumptions of these algorithms, so many online convex MDP algorithms can not apply to the offline setting directly.

---

> ### Author Response · Authors · 2024-08-07
> **References**
>
> [R1]: Aviral Kumar, Joey Hong, Anikait Singh, and Sergey Levine. "Should i run offline reinforcement
> learning or behavioral cloning?" In International Conference on Learning Representations, 2021.
>
> [R2]: Hoang Le, Cameron Voloshin, and Yisong Yue. "Batch policy learning under constraints." In International Conference on Machine Learning, pages 3703–3712. PMLR, 2019.
>
> [R3]: Fan Chen, Junyu Zhang, and Zaiwen Wen. "A near-optimal primal-dual method for off-policy learning in cmdp." Advances in Neural Information Processing Systems, 35:10521–10532, 2022.
>
> [R4]: Kihyuk Hong, Yuhang Li, and Ambuj Tewari. "A primal-dual-critic algorithm for offline constrained reinforcement learning." In International Conference on Artificial Intelligence and Statistics, pages 280–288. PMLR, 2024.
>
> [R5]: Haoran Xu, Xianyuan Zhan, and Xiangyu Zhu. "Constraints penalized q-learning for safe offline reinforcement learning." In Proceedings of the AAAI Conference on Artificial Intelligence, volume 36, pages 8753–8760, 2022.
>
> [R6]: Jongmin Lee, Cosmin Paduraru, Daniel J Mankowitz, Nicolas Heess, Doina Precup, Kee-Eung Kim, and Arthur Guez. "Coptidice: Offline constrained reinforcement learning via stationary distribution correction estimation." In International Conference on Learning Representations, 2021.
>
> [R7]: Aviral Kumar, Justin Fu, Matthew Soh, George Tucker, and Sergey Levine. "Stabilizing off-policy q-learning via bootstrapping error reduction." Advances in neural information processing systems, 32, 2019.
>
> [R8]: Adam Stooke, Joshua Achiam, and Pieter Abbeel. "Responsive safety in reinforcement learning by pid lagrangian methods." In International Conference on Machine Learning, pages 9133–9143. PMLR, 2020.
>
> [R9]: Tom Zahavy, Brendan O’Donoghue, Guillaume Desjardins, and Satinder Singh. "Reward is enough
> for convex mdps." Advances in Neural Information Processing Systems, 34:25746–25759, 2021.

---

> > ### Comment · Reviewer_Za97 · 2024-08-11
> > **I increased my score to 6**
> >
> > Thank the authors for their repsonse and added empirical results. I increased my score to 6.

---

> > > ### Author Response · Authors · 2024-08-11
> > >
> > > We sincerely thank the reviewer for considering our response and revising the score. Much appreciate! Please let us know if you have any follow-up questions. We will be happy to answer them.

---

### Official Review · Reviewer_61FL · 2024-07-14

**Soundness:** 3
**Presentation:** 3
**Contribution:** 2
**Rating:** 5
**Confidence:** 3

**Summary:**

This paper studies the offline safe reinforcement learning (RL) problem and proposes a primal-dual algorithm to solve this problem. The key contributions are that the paper focuses on the more general setting of convex Markov Decision Processes (convex MDPs), where the objective function is a convex utility function, rather than the standard linear rewards. The authors propose a primal-dual algorithm that can handle the offline safe RL problem under the convex MDP setting, crucially requiring only partial data coverage, a weaker assumption compared to previous work. The theoretical analysis provides the sample complexity of the proposed algorithm. Experimental results validate the theoretical findings and demonstrate the practical efficacy of the approach.

**Strengths:**

This paper studies a very interesting and important topic in the RL community. The writing is clear and the results are well presented.

**Weaknesses:**

My main concern with this paper is the limited technical novelty of the contributions. The proposed formulation (8)-(10) appears to be a direct extension of the setup in prior work [26], where the linear reward and cost functions are generalized to convex functions. The algorithm itself is a standard primal-dual method for solving convex optimization problems, rather than a new technical innovation.

Furthermore, the experimental evaluation is quite limited in scope. The authors only compare their approach against a small number of baseline methods, and the experiments are run on a relatively small set of problem instances. This raises questions about the generalizability and practical significance of the empirical results.

While the paper tackles an important and relevant problem in offline safe reinforcement learning, the technical contributions seem incremental given the prior work in this area.

**Questions:**

-- line 181, how large the dataset has to be in order for (8)-(10) to be close to (5)-(7)?

-- When w is parametrized by a neural network, does the proposed formulation satisfy all the assumptions made?

**Limitations:**

see weakness

---

> ### Author Rebuttal · Authors · 2024-08-07
>
> We greatly appreciate the reviewers' detailed comments. Please find our point-by-point response to your questions below.
>
> **Concern of the novelty of the contributions**:  We would like to emphasize that we are the first to study offline convex CMDPs and provide state-of-the-art theoretical results under mild assumptions.
> There are only a few theoretical studies on the offline (linear) CMDPs [R1, R2, R3]. However, these papers either need strong assumptions (full data coverage, Bellman completeness) or can not combine with the general function approximation settings. For this purpose, we develop an efficient algorithm that only needs partial data coverage and meanwhile achieves state-of-the-art theoretical results with general function approximation. The assumption of weak data coverage, the ability to handle large state-action spaces, and the best theoretical results are important aspects that were not addressed in previous works. We would also like to emphasize that the partial data coverage assumption is extremely important in the CMDP setting. A full coverage assumption would mean that each state and action should be covered by the dataset, including the most dangerous states and actions, which is unrealistic and undesirable.
>
> **The proposed formulation appears to be a direct extension of the setup in prior work [26]**: The prior work [26] does not consider the cost function in their formulation. Note that adding the cost function makes the problem more challenging and significantly different from the unconstrained setting. For example, the optimal policy is no longer a greedy policy, as a trade-off needs to be made between reward and costs. This is why CMDP problems are usually much more difficult. In addition, when safety constraints are considered, the gap between program (5)-(7) and approximate program (8)-(10) changes and is unknown in the previous work. Second, extending the program (5)-(7) to program (8)-(10) directly leads to suboptimal results. Finally, the optimal rate of sample complexity is still unknown when considering safety constraints. We focus on answering these questions in our paper.
>
> **The experimental evaluation is limited in the paper**: As our paper is primarily on the theory side, the experiments are used to justify the theoretical results, including the data coverage, function approximation assumption, and the result of sample complexity of our algorithm. According to the comprehensive experiments in [R3] and [R4], CoptiDICE is the most stable and effective algorithm, so we consider CoptiDICE as the baseline in our submission.
>
> But, following the reviewer's suggestions, we conduct a list of experiments in a popular, challenging, and continuous environment--SafetyGym with comprehensive baselines (e.g., CPQ in [R5], PDCA in [R3], CoptiDICE in [R4], and BEAR-Lagrangian in [R6], [R7]). Note that only PDCA and our algorithm provide theoretical results. In these experiments, to deal with the continuous state-action space, we use the fully connected single hidden-layer neural network of width 128 to represent $w$.
>
> We summarize the evaluation results in the following table. All the rewards and costs are normalized. The cost threshold is 1. Each value is averaged over 20 evaluation episodes and 3 random seeds.
>
> | **Task**     | **COptiDICE** |                | **CPQ**    |                | **BEAR-Lag** |                | **PDCA**   |                | **Ours**   |                |
> |--------------|---------------|----------------|------------|----------------|--------------|----------------|------------|----------------|------------|----------------|
> |              | **[Reward, Cost]** |             | **[Reward, Cost]** |           | **[Reward, Cost]** |           | **[Reward, Cost]** |           | **[Reward, Cost]** |           |
> | **AntRun**   | **[0.6, 0.94]** |       | [0.03, 0.02] |      | [0.15, 0.73] |     | [0.28, 0.93] |    | **[0.6, 0.01]** |   |
> | **BallRun**  | [0.59, 3.52] |       | [0.22, 1.27] |      | [-0.47, 5.03] |     | [0.55, 3.38] |    | **[0.24, 0.0]** |   |
> | **BallCircle** | [0.70, 2.61] |     | **[0.64, 0.76]** |      | [0.86, 3.09] |     | [0.63, 2.29] |    | [0.39, 0.93] |  |
> | **CarPush1** | [0.23, 0.5] |        | [-0.03, 0.95] |     | **[0.21, 0.54]** |     | [0.17, 0.41] |   | [0.20, 0.4] |  |
> | **CarRun**   | [0.87, 0.0] |       | [0.95, 1.79] |      | [0.68, 7.78] |     | [0.85, 0.0] |          | **[0.90, 0.0]** |   |
>
> The results show our algorithm is the only one to guarantee safety across all environments. In AntRun, BallRun, and CarRun tasks, our algorithm can achieve the highest reward and small cost among all baseline algorithms.
> Note our algorithm is very flexible to incorporate more sophisticated function approximations (e.g., deeper and more advanced neural networks) to potentially achieve better empirical performance.
>
> **How large the dataset has to be in order for (8)--(10) to be close to (5)--(7)?**: Our theoretical results in Theorem 1 show (8)--(10) will ``converge to'' (5)--(7) with the rate of $\mathcal O(1/\sqrt{n})$, where $n$ is the size of dataset.
>
> **When w is parametrized by a neural network, does the proposed formulation satisfy all the assumptions made?**: Yes. When $w$ is parameterized by a neural network, Assumption 2 (Realizability Assumption) can be satisfied by choosing proper network parameters. In Assumption 3 (Completeness Assumption), the operator $\phi(w)$ or $x$ aims to calculate the $l_1$ norm of constraint function $Kw - (1-\gamma)\mu_0$. The completeness assumption is set up if $l_1$ norm of the constraint function exists, and it works for the case that $w$ is a neural network. In Assumption 4 (Boundness Assumption), if the output of the neural network $w$ is bounded, the assumption is also satisfied.

---

> > ### Comment · Reviewer_61FL · 2024-08-07
> >
> > I would like to thank the authors for their rebuttal, and I have read it. I keep my score and remain positive about this paper.

---

> > > ### Author Response · Authors · 2024-08-08
> > >
> > > Thanks a lot for your precious time in helping us improve our paper. Much appreciated! Please let us know if you have any follow-up questions. We will be happy to answer them.

---

> ### Author Response · Authors · 2024-08-07
> **References**
>
> [R1]: Hoang Le, Cameron Voloshin, and Yisong Yue. "Batch policy learning under constraints." In
> International Conference on Machine Learning, pages 3703–3712. PMLR, 2019.
>
> [R2]: Fan Chen, Junyu Zhang, and Zaiwen Wen. "A near-optimal primal-dual method for off-policy
> learning in cmdp." Advances in Neural Information Processing Systems, 35:10521–10532, 2022.
>
> [R3]: Kihyuk Hong, Yuhang Li, and Ambuj Tewari. "A primal-dual-critic algorithm for offline constrained
> reinforcement learning." In International Conference on Artificial Intelligence and Statistics, pages
> 280–288. PMLR, 2024.
>
> [R4]: Jongmin Lee, Cosmin Paduraru, Daniel J Mankowitz, Nicolas Heess, Doina Precup, Kee-Eung Kim,
> and Arthur Guez. "Coptidice: Offline constrained reinforcement learning via stationary distribution
> correction estimation." In International Conference on Learning Representations, 2021.
>
> [R5]: Haoran Xu, Xianyuan Zhan, and Xiangyu Zhu. "Constraints penalized q-learning for safe offline
> reinforcement learning." In Proceedings of the AAAI Conference on Artificial Intelligence, volume 36,
> pages 8753–8760, 2022.
>
> [R6]: Aviral Kumar, Justin Fu, Matthew Soh, George Tucker, and Sergey Levine. "Stabilizing off-policy
> q-learning via bootstrapping error reduction." Advances in neural information processing systems,
> 32, 2019.
>
> [R7]: Adam Stooke, Joshua Achiam, and Pieter Abbeel. "Responsive safety in reinforcement learning
> by pid lagrangian methods." In International Conference on Machine Learning, pages 9133–9143.
> PMLR, 2020.

---

### Decision · Program_Chairs · 2024-09-25

**Decision:**

Accept (poster)

**Comment:**

This work on constrained MDPs studies sample complexity under general function approximation, partial coverage, and general (convex) objective functionals. The proposed primal-dual approach achieves better sample complexity w.r.t. the state of the art, and under more general conditions.

The reviewers questioned the technical novelty of the contribution and the limited scope of the experiments.
However, during the discussion phase, the authors provided additional experiments on different environments, and clarified what are the improvements over the state of the art and the technical challenges that were involved.

For this reason, I recommend acceptance of this paper, conditional on the inclusion of the new experiments in the camera ready version, and the implementation of the other improvements suggested by the reviewers, especially regarding presentation. I also suggest to add to the manuscript a discussion on the technical novelty, in order to prevent further misunderstandings.